# EMERGENCE OF MAPS
# IN THE MEMORIES
# OF BLIND NAVIGATION AGENTS

**Erik Wijmans**[1,2][*] **Manolis Savva**[2,3] **Irfan Essa**[1,4] **Stefan Lee**[5] **Ari S. Morcos**[2] **Dhruv Batra**[1,2]
[1]Georgia Institute of Technology   [2]FAIR, Meta AI   [3]Simon Fraser University
[4]Google Research Atlanta [5]Oregon State University

## ABSTRACT

Animal navigation research posits that organisms build and maintain internal spatial representations, or maps, of their environment. We ask if machines – specifically, artificial intelligence (AI) navigation agents – also build implicit (or 'mental') maps. A positive answer to this question would (a) explain the surprising phenomenon in recent literature of ostensibly map-free neural-networks achieving strong performance, and (b) strengthen the evidence of mapping as a fundamental mechanism for navigation by intelligent embodied agents, whether they be biological or artificial. Unlike animal navigation, we can judiciously design the agent's perceptual system and control the learning paradigm to nullify alternative navigation mechanisms. Specifically, we train 'blind' agents – with sensing limited to only egomotion and *no other sensing of any kind* – to perform PointGoal navigation ('*go to $\Delta x$, $\Delta y$*') via reinforcement learning. Our agents are composed of navigation-agnostic components (fully-connected and recurrent neural networks), and our experimental setup provides *no inductive bias towards mapping*. Despite these harsh conditions, we find that blind agents are (1) surprisingly effective navigators in *new* environments ($\sim$95% success); (2) they utilize memory over long horizons (remembering $\sim$1,000 steps of past experience in an episode); (3) this memory enables them to exhibit intelligent behavior (following walls, detecting collisions, taking shortcuts); (4) there is *emergence of maps* and *collision detection neurons* in the representations of the environment built by a blind agent as it navigates; and (5) the emergent maps are selective and task dependent (*e.g.* the agent 'forgets' exploratory detours). Overall, this paper presents no new techniques for the AI audience, but a surprising finding, an insight, and an explanation.

## 1   INTRODUCTION

Decades of research into intelligent animal navigation posits that organisms build and maintain internal spatial representations (or maps)[1] of their environment, that enables the organism to determine and follow task-appropriate paths (Tolman, 1948; O'keefe & Nadel, 1978; Epstein et al., 2017). Hamsters, wolves, chimpanzees, and bats leverage prior exploration to determine and follow shortcuts they may never have taken before (Chapuis & Scardigli, 1993; Peters, 1976; Menzel, 1973; Toledo et al., 2020; Harten et al., 2020). Even blind mole rats and animals rendered situationally-blind in dark environments demonstrate shortcut behaviors (Avni et al., 2008; Kimchi et al., 2004; Maaswinkel & Whishaw, 1999). Ants forage for food along meandering paths but take near-optimal return trips (Müller & Wehner, 1988), though there is some controversy about whether insects like ants and bees are capable of forming maps (Cruse & Wehner, 2011; Cheung et al., 2014).

Analogously, mapping and localization techniques have long played a central role in enabling non-biological navigation agents (or robots) to exhibit intelligent behavior (Thrun et al., 2005; Institute,

---

[*]Correspondence to etw@gatech.edu.

[1]Throughout this work, we use 'maps' to refer to a spatial representation of the environment that enables intelligent navigation behavior like taking shortcuts. We provide a detailed discussion and contrast w.r.t. a 'cognitive map' as defined by O'keefe & Nadel (1978) in Apx. B.1.

1972; Ayache & Faugeras, 1988; Smith et al., 1990). More recently, the machine learning community has produced a surprising phenomenon – neural-network models for navigation that curiously do not contain any explicit mapping modules but still achieve remarkably high performance (Savva et al., 2019; Wijmans et al., 2020; Kadian et al., 2020; Chattopadhyay et al., 2021; Khandelwal et al., 2022; Partsey et al., 2022; Reed et al., 2022). For instance, Wijmans et al. (2020) showed that a simple 'pixels-to-actions' architecture (using a CNN and RNN) can navigate to a given point in a novel environment with near-perfect accuracy; Partsey et al. (2022) further generalized this result to more realistic sensors and actuators. Reed et al. (2022) showed a similar general purpose architecture (a transformer) can perform a wide variety of embodied tasks, including navigation. The mechanisms explaining this ability remain unknown. Understanding them is both of scientific and practical importance due to safety considerations involved with deploying such systems.

In this work, we investigate the following question – is mapping an emergent phenomenon? Specifically, do artificial intelligence (AI) agents learn to build internal spatial representations (or 'mental' maps) of their environment as a natural consequence of learning to navigate?

The specific task we study is PointGoal navigation (Anderson et al., 2018), where an AI agent is introduced into a new (unexplored) environment and tasked with navigating to a relative location – '*go 5m north, 2m west relative to start*'[2]. This is analogous to the direction and distance of foraging locations communicated by the waggle dance of honey bees (Von Frisch, 1967).

Unlike animal navigation studies, experiments with AI agents allow us to precisely isolate mapping from alternative mechanisms proposed for animal navigation – the use of visual landmarks (Von Frisch, 1967), orientation by the arrangement of stars (Lockley, 1967), gradients of olfaction or other senses (Ioalè et al., 1990). We achieve this isolation by judiciously designing the agent's perceptual system and the learning paradigm such that these alternative mechanisms are rendered implausible. Our agents are effectively 'blind'; they possess a *minimal perceptual system* capable of sensing *only egomotion*, *i.e.* change in the agent's location and orientation as the it moves – no vision, no audio, no olfactory, no haptic, no magnetic, or any other sensing of any kind. This perceptual system is deliberately impoverished to isolate the contribution of memory, and is inspired by blind mole rats, who perform localization via path integration and use the Earth's magnetic field as a compass (Kimchi et al., 2004). Further still, our agents are composed of navigation-agnostic, generic, and ubiquitous architectural components (fully-connected layers and LSTM-based recurrent neural networks), and our experimental setup provides *no inductive bias towards mapping* – no map-like or spatial structural components in the agent, no mapping supervision, no auxiliary tasks, nothing other than a reward for making progress towards a goal.

Surprisingly, even under these deliberately harsh conditions, we find the emergence of map-like spatial representations in the agent's non-spatial unstructured memory, enabling it to not only successfully navigate to the goal but also exhibit intelligent behavior (like taking shortcuts, following walls, detecting collisions) similar to aforementioned animal studies, and predict free-space in the environment. Essentially, we demonstrate an 'existence proof' or an ontogenetic developmental account for the emergence of mapping without any previous predisposition. Our results also explain the aforementioned surprising finding in recent literature – that ostensibly map-free neural-network achieve strong autonomous navigation performance – by demonstrating that these 'map-free' systems in fact learn to construct and maintain map-like representations of their environment.

Concretely, we ask and answer following questions:

1) *Is it possible to effectively navigate with just egomotion sensing?* Yes. We find that our 'blind' agents are *highly* effective in navigating *new* environments – reaching the goal with $95.1\%_{\pm1.3\%}$ success rate. And they traverse moderately efficient (though far from optimal) paths, reaching $62.9\%_{\pm1.6\%}$ of optimal path efficiency. We stress that these are novel testing environments, the agent has not memorized paths within a training environment but has learned efficient navigation strategies that generalize to novel environments, such as emergent wall-following behavior.

2) *What mechanism explains this strong performance by 'blind' agents?* Memory. We find that memoryless agents completely fail at this task, achieving nearly 0% success. More importantly, we find that agents with memory utilize information stored over a long temporal and spatial horizon and that collision-detection neurons emerge within this memory. Navigation performance as a function of the number of past actions/observations encoded in the agent's memory does not

---

[2]The description in English is purely for explanatory purposes; the agent receives relative goal coordinates.

saturate till one thousand steps (corresponding to the agent traversing $89.1_{\pm 0.66}$ meters), suggesting that the agent 'remembers' a long history of the episode.

3) *What information does the memory encode about the environment?* Implicit maps. We perform an AI rendition of Menzel (1973)'s experiments, where a chimpanzee is carried by a human and shown the location of food hidden in the environment. When the animal is set free to collect the food, it does not retrace the demonstrator's steps but takes shortcuts to collect the food faster. Analogously, we train a blind agent to navigate from a source location (**S**) to a target location (**T**). After it has finished navigating, we transplant its constructed episodic memory into a second 'probe'-agent (which is also blind). We find that this implanted-memory probe-agent performs *dramatically better* in navigating from **S** to **T** (and **T** to **S**) than it would without the memory transplant. Similar to the chimpanzee, the probe agent takes shortcuts, typically cutting out backtracks or excursions that the memory-creator had undertaken as it tried to work its way around the obstacles. These experiments provide compelling evidence that blind agents learn to build and use implicit map-like representations of their environment solely through learning to navigate. Intriguingly further still, we find that surprisingly detailed *metric occupancy maps* of the environment (indicating free-space) can be explicitly decoded from the agent's memory.

4) *Are maps task-dependent?* Yes. We find that the emergent maps are a function of the navigation goal. Agents 'forget' excursions and detours, *i.e.* their episodic memory only preserves the features of the environment relevant to navigating to their goal. This, in part, explains why transplanting episodic memory from one agent to another leads it to take shortcuts – because the excursion and detours are simply forgotten.

Overall, our experiments and analyses demonstrate that 'blind' agents solve PointGoalNav by combining information over long time horizons to build detailed maps of their environment, solely through the learning signals imposed by goal-driven navigation. In biological systems, convergent evolution of analogous structures that cannot be attributed to a common ancestor (*e.g.* eyes in vertebrates and jellyfish (Kozmik et al., 2008)) is often an indicator that the structure is a *natural response* to the ecological niche and selection pressures. Analogously, our results suggest that mapping may be a *natural solution* to the problem of navigation by intelligent embodied agents, whether they be biological or artificial. We now describe our findings for each question in detail.

## 2 BLIND AGENTS ARE EFFECTIVE NAVIGATORS

We train navigation agents for PointGoalNav in virtualized 3D replicas of real houses utilizing the AI Habitat simulator (Savva et al., 2019; Szot et al., 2021) and Gibson (Xia et al., 2018) and Matterport3D (Chang et al., 2017) datasets. The agent is physically embodied as an cylinder with a diameter $0.2$m and height $1.5$m. In each episode, the agent is randomly initialized in the environment, which establishes an episodic agent-centric coordinate system. The goal location is specified in cartesian coordinates $(x_g, y_g, z_g)$ in this system. The agent has four actions – move_forward ($0.25$ meters), turn_left ($10°$), turn_right ($10°$), and stop (to signal reaching the goal), and allowed a maximum of 2,000 steps to reach the specified goal. It is equipped with an egomotion sensor providing it relative position $(\Delta x, \Delta y, \Delta z)$ and relative 'heading' (or yaw angle) $\Delta\theta$ between successive steps, which is integrated to track the agent's pose relative to start $[x_t, y_t, z_t, \theta_t]$. This is sometimes referred to as a 'GPS+Compass' sensor in this literature (Savva et al., 2019; Wijmans et al., 2020).

We use two task-performance dependent metrics: i) Success, defined as whether the agent predicted stop within 0.2 meters of the target, and ii) Success weighted by inverse Path Length (SPL) (Anderson et al., 2018), defined as success weighted by the efficiency of agent's path compared to the oracle path (the shortest path). Given the high success rates we observe, SPL can be roughly interpreted as efficiency of the path taken compared to the oracle path – *e.g.* 95% SPL means the agent took a path 95% as efficient as the oracle path while 50% SPL means the agent was 50% as efficient. Performance is evaluated in previously *unseen* environments to test generalization, not just memorization.

The agent's policy is instantiated as a long short-term memory (LSTM) (Hochreiter & Schmidhuber, 1997) recurrent neural network – formally, given current observations $\mathbf{o}_t = [x_g, y_g, z_g, x_t, y_t, z_t, \theta_t]$, $(\mathbf{h}_t, \mathbf{c}_t) = \text{LSTM}(\mathbf{o}_t, (\mathbf{h}_{t-1}, \mathbf{c}_{t-1}))$. We refer to this $(\mathbf{h}_t, \mathbf{c}_t)$ as the agent's internal memory representation. Note that only contains information gathered during the current navigation episode. We train our agents for this task using a reinforcement learning (Sutton & Barto, 1992) algorithm called DD-PPO (Wijmans et al., 2020). The reward has a term for making progress towards the goal and for successfully reaching it. Neither the training procedure nor agent architecture contain explicit inductive biases towards mapping or planning relative to a map. Apx. A.1 describes training details.

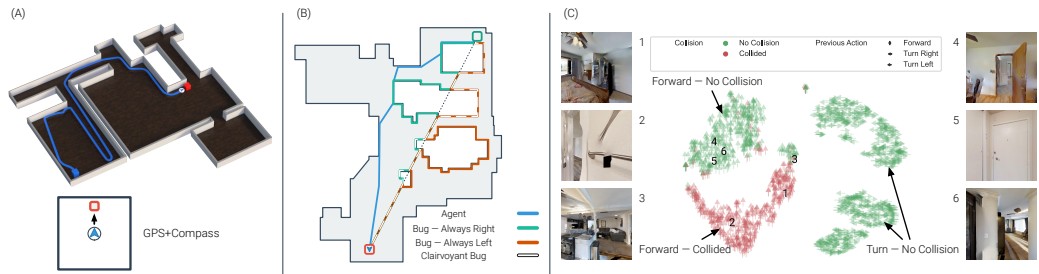

Figure 1: **(A) PointGoal navigation**. An agent is initialized in a novel environment (blue square) and task with navigation to a point specified relative to the start location (red square). We study 'blind' agents, equipped with just an egomotion sensor (called GPS+Compass in this literature). **(B) 'Blind' agent vs. bug.** Our learned 'blind' agent compared to 2 variants and an oracle equipped variant of the Bug algorithm (Lumelsky & Stepanov, 1987). The Bug algorithm initially orients itself towards the goal and then proceeds towards the goal. Upon hitting a wall, it follows along the wall until it reaches the other side. The oracle version is told whether wall-following left or right is optimal, providing an upper-bound on Bug algorithm performance. **(C) t-SNE of the agent's internal representation for collisions.** We find 4 overall clusters corresponding to the previous action taken and whether or not that action led to a collision.

Surprisingly, we find that agents with this impoverished sensing regime are able to navigate with *near-perfect efficacy* – reaching the goal with $95.1\%_{\pm1.3\%}$ success rate (Table 1), even when the agent must take *hundreds* of actions and traverse over 25m. This performance is similar in success rate (95.1 vs 94.0)[3] to a *sighted* agent (equipped with a depth camera) trained on a larger dataset (HM3D) (Ramakrishnan et al., 2021). The paths taken by the blind agent are moderately efficient but (as one might expect) far less so than a sighted agent (62.9 vs 83.0 SPL).

At this point, it might be tempting to believe that this is an easy navigation problem, but we urge the reader to fight hindsight bias. We contend that the SPL of

| | Agent | Success | SPL |
|---|---|---|---|
| 1 | Blind | $95.1_{\pm1.3}$ | $62.9_{\pm1.6}$ |
| 2 | Clairvoyant Bug | $100_{\pm0.0}$ | 46.0 |
| 3 | Sighted (Depth) (Ramakrishnan et al., 2021) | 94.0 | 83.0 |

Table 1: **PointGoalNav performance** agents on PointGoalNav. We find that blind agents are surprisingly effective (success) though not efficient (SPL) navigators. They have similar success as an agent equipped with a Depth camera and higher SPL than a clairvoyant version of the 'Bug' algorithm.

this blind agent is surprisingly high given the impoverished sensor suite. To put this SPL in context, we compare it with 'Bug algorithms' (Lumelsky & Stepanov, 1987), which are motion planning algorithms inspired by insect navigation, involving an agent equipped with only a localization sensor. In these algorithms, the agent first orients itself towards the goal and then travels directly towards it until it encounters a wall, in which case it follows along the wall along one of two directions of travel. The primary challenge for Bug algorithms is determining whether to go left or right upon reaching a wall. To provide an upper bound on performance, we implement a 'clairvoyant' Bug algorithm agent with an oracle that tells it whether left or right is optimal. Even with the additional privileged information, the 'clairvoyant' Bug agent achieves an SPL of 46%, which is considerably less efficient than the 'blind' agent. Fig. 1b shows an example of the path our blind agent takes compared to 3 variants of the Bug algorithm. This shows that blind navigation agents trained with reinforcement learning are highly efficient at navigating in previously unseen environments given their sensor suite.

## 2.1 Emergence of wall-following behavior and collision-detection neurons

Fig. 1b shows the blind agent exhibiting wall-following behavior (also see blue paths in Fig. A6 and videos in supplement). This behavior is remarkably consistent; the agent spends the majority of an episode near a wall. This is surprising because it is trained to navigate as *quickly* as possible, thus, it would be rewarded for traveling in straighter paths (that avoid walls). We hypothesize that this strategy emerges due to two factors. 1) The agent is blind, it has no way to determine where obstacles are besides 'bumping' into them. 2) The environment is unknown to the agent. While this is clearly true for testing environments it is also *functionally* true for training environments because the coordinate

---

[3]It may seem like the blind agent outperforms the sighted agent, but performance is within our error bars.

system is *episodic*, every episode uses a randomly-instantiated coordinate system based on how the agent was spawned; and the since the agent is blind, it cannot perform visual localization.

We test both hypotheses. To test (2), we provide an experiment in Apx. C.1 showing that when the agent is trained in a single environment with a consistent global coordinate system, it learns to memorize the shortest paths in this environment and wall-following does *not* emerge. Consequently, this agent is unable to navigate in new environment, achieving 100% success on train and 0% on test.

To test (1), we analyze whether the agent is capable of detecting collisions. Note that the agent is not equipped with a collision sensor. In principle, the agent can infer whether it collided – if tries to move forward and the resulting egomotion is *atypical*, then it is likely that a collision happened. This leads us to ask – *does the agent's memory contain information about collisions?* We train a linear classifier that uses the (frozen) internal representation $(\mathbf{h}_{t+1}, \mathbf{c}_{t+1})$ to predict if action $a_t$ resulted in a collision (details in Apx. A.5). The classifier achieves 98% accuracy on held-out data. As comparison, random guessing on this 2-class problem would achieve 50%. This shows the agent's memory not only predicts its collisions, but also that *collision-vs-not* are linearly separable in internal-representation space, which strongly suggests that the agent has learned a collision sensor.

Next, we examine how collisions are structured in the agent's internal representation by identifying the subspace that is used for collisions. Specifically, we re-train the linear classifier with an $\ell_1$-weight penalty to encourage sparsity. We then select the top 10 neurons (from 3072) with the largest weight magnitude; this reduces dimensionality by 99.7% while still achieving 96% collision-vs-not accuracy. We use t-SNE (Van der Maaten & Hinton, 2008) to create a 2-dimension visualization. We find 4 distinct semantically-meaningful clusters (Fig. 1c). One cluster always fires for collisions, one for forward actions that did not result in a collision, and the other two correspond to turning actions. Notice that these exceedingly small number of dimensions and neurons essentially predict *all* collisions and movement of the agent. We include videos in the supplementary materials.

## 3 MEMORY IS USED OVER LONG HORIZONS

Next, we examine how memory is utilized by asking if the agent uses memory solely to remember short-term information (*e.g.* did it collide in the last step?) or whether it also includes long-range information (*e.g.* did it collide hundreds of steps ago?). To answer this question, we restrict the memory capacity of our agent. Specifically, let $k$ denote the memory budget. At each time $t$, we take the previous $k$ observations, $[o_{t-k+1}, \ldots, o_t]$, and construct the internal representation $(\mathbf{h}_t, \mathbf{c}_t)$ via the recurrence $(\mathbf{h}_i, \mathbf{c}_i) = \text{LSTM}(o_i, (\mathbf{h}_{i-1}, \mathbf{c}_{i-1}))$ for $t - k < i \leq t$ where $(\mathbf{h}_{t-k}, \mathbf{c}_{t-k}) = (\mathbf{0}, \mathbf{0})$.

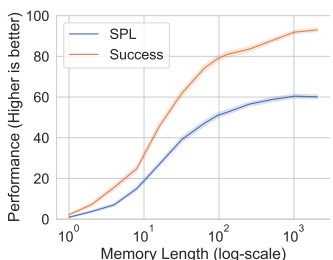

Figure 2: **Navigation performance vs. memory length.** Agent performance does not saturate until memory can contain information from hundreds of steps. A memory of $10^3$ steps is half the maximum episode length.

If the agent is only leveraging its memory for short-term storage we would expect performance to saturate at a small value of $k$. Instead, Fig. 2 shows that the agent leverages its memory for *significantly* long term storage. When memoryless ($k = 1$), the agent completely fail at the task, achieving nearly 0% success. Navigation performance as a function of the memory budget ($k$) does not saturate till *one thousand steps*. Recall that the agent can move forward 0.25 meters or turn 10° at each step. The average distance traveled in 1000 steps is 89.1$_{\pm 0.66}$ meters, indicating that it remembers information over long temporal and spatial horizons. In Apx. C.6 we train agents to operate at a specific memory budget. We find that a budget of $k = 256$, the largest we are able to train, is not sufficient to achieve the performance of unbounded.

## 4 MEMORY ENABLES SHORTCUTS

To investigate what information is encoded in the memory of our blind agents, we develop an experimental paradigm based on 'probe' agents. A probe is a secondary navigation agent[4] that is structurally identical to the original (sensing, architecture, *etc.*), but parametrically augmented with the primary agent's constructed episodic memory representation $(\mathbf{h}_T, \mathbf{c}_T)$. The probe has no influence

---

[4]To avoid confusion, we refer to this probe agent as 'probe' and the primary agent as 'agent' from this point.

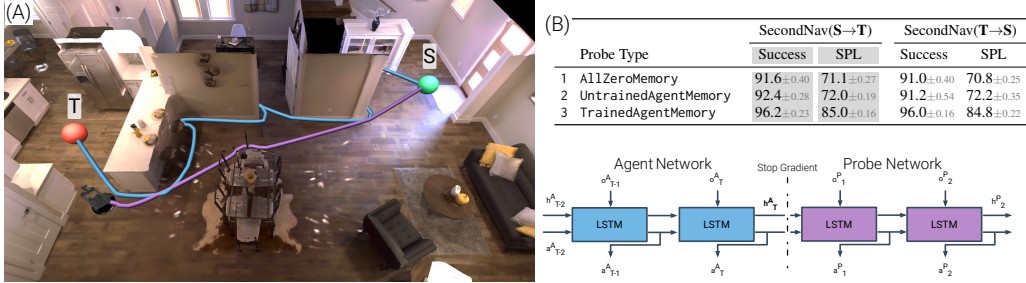

Figure 3: **(A) Probe experiment.** First, an agent navigates (blue path, blue LSTM) from start (green sphere) to target (red sphere). After the agent navigates, we task a probe (purple LSTM) with performing the same navigation episode with the additional information encapsulated in the agent's internal representation (or memory), $\mathbf{h_T^A}$. The probe is able to navigate more efficiently by taking shortcuts (purple path). As denoted by the dashed line between the probe and agent networks, the probe does not influence what the agent stores in its internal representation. **(B) Agent memory increases probe efficiency (SPL).** Results of our trained probe agent under three configurations – initialized with an empty representation (AllZeroMemory), a representation of a random agent walked along the trained agent's path (UntrainedAgentMemory), and the final representation of the trained agent (TrainedAgentMemory). 95% confidence interval reported over 5 agent-probe pairs.

on the agent, *i.e.* no gradients (or rewards) follow from probe to agent (please see training details in Apx. A.2). We use this paradigm to examine whether the agent's final internal representation contains sufficient information for taking shortcuts in the environment.

As illustrated in Fig. 3A, the agent first navigates from source ($\mathbf{S}$) to target ($\mathbf{T}$). After the agent reaches $\mathbf{T}$, a probe is initialized[5] at $\mathbf{S}$, its memory initialized with the agent's final memory representation, *i.e.* $(\mathbf{h}_0, \mathbf{c}_0)^{\text{probe}} = (\mathbf{h}_T, \mathbf{c}_T)^{\text{agent}}$, and tasked with navigating to $\mathbf{T}$. We refer to this probe task as SecondNav($\mathbf{S}{\rightarrow}\mathbf{T}$). All evaluations are conducted in environments not used for training the agent nor the probe. Thus, any environmental information in the agent's memory must have been gathered during its trajectory (and not during any past exposure during learning). Similarly, all initial knowledge the probe has of the environment must come from the agent's memory $(\mathbf{h}_T, \mathbf{c}_T)^{\text{agent}}$.

Our hypothesis is that the agent's memory contains a spatial representation of the environment, which the probe can leverage. If the hypothesis is true, we would expect the probe to navigate SecondNav($\mathbf{S}{\rightarrow}\mathbf{T}$) more efficiently than the agent (*e.g.* by taking shortcuts and cutting out exploratory excursions taken by the agent). If not, we would expect the probe to perform on-par with the agent since the probe is being trained on essentially the same task as the agent[6]. In our experiments, we find that the probe is *significantly* more efficient than the agent – SPL of 62.9%$_{\pm 1.6\%}$ (agent) vs. 85.0%$_{\pm 0.16\%}$ (probe). It is worth stressing how remarkable the performance of the probe is – in a new environment, a blind probe navigating without a map traverses a path that is within 15% of the *shortest path* on the map. The best known *sighted* agents (equipped with an RGB camera, Depth sensor, and egomotion sensor) achieve an SPL of 84% on this task (Ramakrishnan et al., 2021). Essentially, the memories of a blind agent are as valuable as having vision!

Fig. 3A shows the difference in paths between the agent and probe (and videos showing more examples are available in the supplement). While the agent exhibits wall-following behavior, the probe instead takes more direct paths and rarely performs wall following. Recall that the only difference in the agent and probe is the contents of the initial hidden state – reward is identical (and available only during training), training environments are identical (although the episodes are different), and evaluation episodes are identical – meaning that the environmental representation in the agent's episodic memory is what enables the probe to navigate more efficiently.

We further compare this result (which we denote as TrainedAgentMemory) with two control groups: 1) AllZeroMemory: An empty (all zeros) episodic memory to test for any systematic biases in the

---

[5]The probe's heading at $\mathbf{S}$ is set to the agent's final heading upon reaching $\mathbf{T}$.

[6]We note that an argument can be made that if the agent's memory is useless to the probe, then the probe is being trained on a *harder* task since it must learn to navigate and ignore the agent's memory. But this argument would predict the probe's performance to be *lower* not higher than the agent.

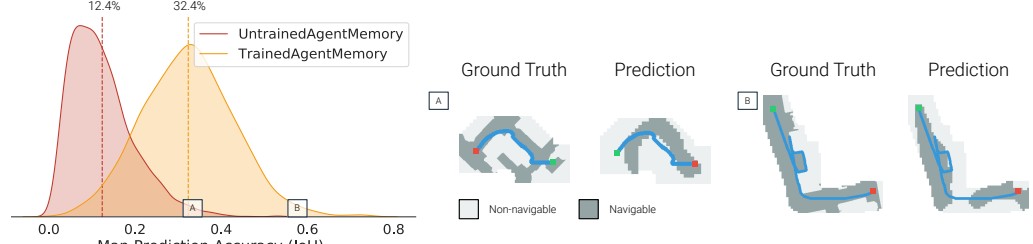

Figure 4: **Learning navigation improves map prediction.** (Left) Accuracy (Intersection over Union) distributions (kernel density estimation) and means (dashed lines); `TrainedAgentMemory` has a higher mean than `UntrainedAgentMemory` with p-value $\leq 10^{-5}$ (via Wilcoxon (1992) signed-rank test). (Right) Example ground truth and predicted occupancy maps for `TrainedAgentMemory` (corresponding to (A) and (B) IoU points). Light grey is non-navigable and dark grey is navigable. The agent (light blue path) navigates from start (green) to target (red). Notice that when the agent travels close to a wall, the decoder predicts another parallel to it, indicating a corridor.

probe tasks. This probe contains identical information at the start of an episode as the agent (*i.e.* no information). 2) `UntrainedAgentMemory`: Episodic memory generated by an untrained agent (*i.e.* with random parameters) as it is walked along the trajectory of the trained agent. This disentangles the agent's structure from its parameters; and tests whether simply being encoded by an LSTM (even one with random parameters) provides an inductive bias towards building good environmental representations (Wieting & Kiela, 2019). We find no evidence for this bias – `UntrainedAgentMemory` performs no better than `AllZeroMemory` (Fig. 3B, row 1 vs. 2). Furthermore, `TrainedAgentMemory` significantly outperforms both controls by $+13$ points SPL and $+4$ points Success (Fig. 3B, row 3 vs. 1 and 2). Taken together, these two results indicate that the ability to construct useful spatial representations of the environment from a trajectory is decidedly a learned behavior.

Next, we examine if there is any directional preference in the episodic memory constructed by the agent. Our claim is that even though the agent navigates from $\mathbf{S}$ to $\mathbf{T}$, if its memory indeed contains map-like spatial representations, it should also support probes for the reverse task Second-Nav($\mathbf{T} \rightarrow \mathbf{S}$). Indeed, we find that `TrainedAgentMemory` probe performs the same (within margin of error) on both SecondNav($\mathbf{S} \rightarrow \mathbf{T}$) and SecondNav($\mathbf{T} \rightarrow \mathbf{S}$) (Fig. 3B right column) – indicating that the memory is equally useful in both directions. In Apx. C.2 we demonstrate that the probe removes excursions from the agent's path and takes shortcuts through previously unseen parts of the environment. Overall, these results provide compelling evidence that blind agents learn to build and use implicit map-like representations that enable shortcuts and reasoning about previously *untraversed* locations in the environment, solely through learning to navigate between two points.

## 5 LEARNING NAVIGATION IMPROVES METRIC MAP DECODING

Next, we tackle the question *'Does the agent build episodic representations capable of decoding metric maps (occupancy grids) of the environment?'*. Formally, given the final representation $(\mathbf{h}_T, \mathbf{c}_T)^{\text{agent}}$, we train a separate decoding network to predict an allocentric top-down occupancy grid (free-space vs not) of the environment. As with the probes, no gradients are propagated from the decoder to the agent's internal representation. We constrain the network to make predictions for a location only if the agent reached within 2.5 meters of it (refer to Apx. A.3 for details). Note that since the agents are 'blind' predictions about *any* unvisited location require reasoning about unseen space. As before, we compare the internal representation produced by `TrainedAgentMemory` to internal representation produced by an agent with random parameters, `UntrainedAgentMemory`.

Fig. 4 shows the distribution of map-prediction accuracy, measured as interaction-over-union (IoU) with the true occupancy grid. We find that `TrainedAgentMemory` enables uniformly more accurate predictions than `UntrainedAgentMemory`– 32.5% vs 12.5% average IoU. The qualitative examples show that the predictor is commonly able to make accurate predictions about unvisited locations, *e.g.* when the agent travels close to one wall, the decoder predicts another parallel to it, indicating a corridor. These results show that the internal representation contains necessary information to decode accurate occupancy maps, even for unseen locations. We note that the environment structural priors are also necessary to prediction unseen locations. Thus agent memory is necessary but not sufficient.

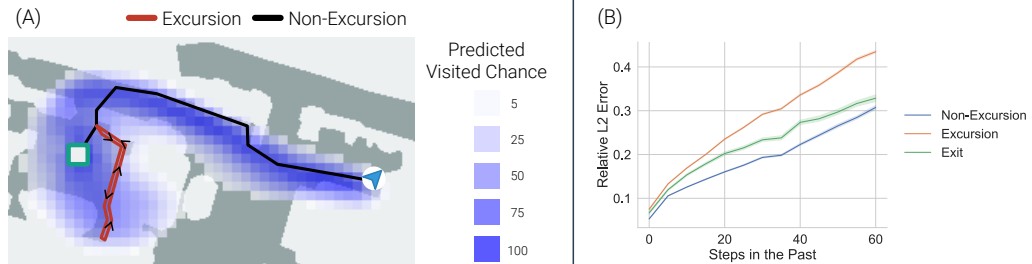

Figure 5: **(A) Excursion prediction example.** Qualitative example of the previously-visited location decoder making systematic errors. Blue represents the confidence of the decoder that the agent was previously at a given location; we can see that it is lower in the path interval marked in red (excursion) than the rest. **(B) Remembrance of excursions.** Performance broken down into three categories. *'Non-excursion'* is all predictions where the current location of the agent and the prediction time step are not part of an excursions. *'Excursion'* is when the prediction time step is part of an excursion. *'Exit'* is when the prediction time step is part of the last 10% of the excursion. X-axis is the distance into the past and Y-axis is the relative error between the true and predicted locations.

In Apx. C.4, we conduct this analysis on 'sighted' navigation agents (equipped with a `Depth` camera and egomotion sensor). Perhaps counter-intuitively, we do not find conclusive evidence that metric maps can be decoded from the memory of sighted agents (despite their sensing suite being a strict superset of blind agents). Our conjecture is that for higher-level strategies like map-building to emerge, the learning problem must not admit 'trivial' solutions such as the ones deep reinforcement learning is know to latch onto (Baker et al., 2020; Lehman et al., 2020; Kadian et al., 2020). We believe that the minimal perception system used in our work served to create a challenging learning problem, which in turn limited the possible 'trivial' solutions, thus inducing map-building.

## 6 MAPPING IS TASK-DEPENDENT: AGENT FORGETS EXCURSIONS

Given that the agent is memory-limited, it stands to reason that it might need to choose what information to preserve and what to 'forget'. To examine this, we attempt to decode the agent's past positions from its memory. Formally, given internal state at time $t$, $(\mathbf{h}_t, \mathbf{c}_t)$, we train a network $f_k(\cdot)$ to predict the agent's location $k$ steps in to the past, *i.e.* $\hat{s}_{t-k} = f_k(\mathbf{h}_t, \mathbf{c}_t) + s_t$, $k \in [1, 256]$. Given ground truth location $s_{t-k}$, we evaluate the decoder via relative L2 error $||\hat{s}_{t-k} - s_{t-k}||/||s_{t-k} - s_t||$ (see Apx. A.4 for details). Qualitative analysis shows that the agent forgets excursions[7], *i.e.* excursions are harder to decode (see Fig. 5a). To quantify this, we manually labelled excursions in 216 randomly sampled episodes in evaluation environments. Fig. 5b shows that excursions are harder to decode than non-excursions, indicating that the agent does indeed forget excursions. Interestingly, we find that the exit of the excursion is considerably easier to decode, indicating that the end of the excursion performs a similar function to landmarks in animal and human navigation (Chan et al., 2012).

In the appendix, we study several additional questions that could not be accommodated in the main paper. In Apx. C.2 we further examine the probe's performance. In Apx. C.3 we examine predicting future agent locations. In Apx. C.5 we use agent's hidden state as a world model.

## 7 RELATED WORK

**Spatial representations.** Prior work has shown that LSTMs build grid-cell (O'keefe & Nadel, 1978) representations of an environment when trained for path integration within that environment (Banino et al., 2018; Cueva & Wei, 2018; Sorscher et al., 2020). In contrast, our work provides no direct supervision for path integration, localization, or mapping. Banino et al. (2018) demonstrated that these maps aid in navigation by training a navigation agent that utilizes this cognitive map. In contrast, we show that LSTMs trained for navigation learn to build spatial representations in novel environments. Whether or not LSTMs trained under this setting also utilize grid-cells is a question for future work. Bruce et al. (2018) demonstrated that LSTMs learn localization when trained for navigation in a single environment. We show that they learn mapping when given location and trained in many environments. Dwivedi et al. (2022) examined what navigation agents learn about

---

[7]We define an excursion as a sub-path that approximately forms a loop.

their environments. We provided a detailed account of emergent mapping in larger environments, over longer time horizons, and show the emergence of intelligent behavior and mapping in blind agents, which is not the focus of prior work.

Graves et al. (2016) proposed a differentiable neural computer and showed that spatial representations emerge from graph tasks and block puzzles. Huynh et al. (2020) proposed a spatial memory architecture and show that a map-like representation emerges when trained for localization. We show that spatial representations emerge in task-agnostic neural networks trained for navigation.

**'Map-free' navigation agents.** Learned agents that navigate without an explicit mapping module (called 'map-free' or 'pixels-to-actions') have shown strong performance on a variety of tasks (Savva et al., 2019; Wijmans et al., 2020; Kadian et al., 2020; Chattopadhyay et al., 2021; Khandelwal et al., 2022; Partsey et al., 2022; Reed et al., 2022). In this work, we do not provide any novel techniques nor make any experimental advancement in the efficacy of such (sighted) agents. However, we make two key findings. First, that blind agents are highly effective navigators for `PointGoalNav`, exhibiting similar efficacy as sighted agents. Second, we begin to explain how 'map-free' navigation agents perform their task: they build implicit maps in their memory, although the story is a bit nuanced due to the results in Apx. C.4; we suspect this understanding might be extended in future work.

## 8 OUTLOOK: LIMITATIONS, REPRODUCIBILITY

In this work, we have shown that 'blind' AI navigation agents – agents with similar perception as blind mole rats – are capable of performing goal-driven navigation to a high degree of performance. We then showed that these AI navigation agents learn to build map-like representations (supporting the ability to take shortcuts, follow walls, and predict free-space and collisions) of their environment solely through learning goal-driven navigation. Our agents and training regime have no added inductive bias towards map-building, be it explicit or implicit, implying that cognitive maps may be a *natural solution* to the inductive biases imposed by navigation by intelligent embodied agents, whether they be biological or artificial. In a similar manner, convergent evolution (Kozmik et al., 2008), where two unrelated intelligent systems independently arrive at similar mechanisms, suggests that the mechanism is a natural response of having to adapt to the environment and the task.

Our results also provide an explanation of the surprising success of map-free neural network navigation agents by showing that these agents in fact learn to build map-like internal representations with no learning signal other than goal driven navigation. This result establish a link between how 'map-free' systems navigate with analytic mapping-and-planning techniques (Thrun et al., 2005; Institute, 1972; Ayache & Faugeras, 1988; Smith et al., 1990).

Our results and analyses also point towards future directions in AI navigation research. Specifically, imbuing AI navigation agents with explicit (*e.g.* architectural design) or implicit (*e.g.* training regime or auxiliary objectives) priors that bias agents towards learning an internal representation with the features found here may improve their performance. Further, it may better equip them to learn more challenging tasks such as rearrangement of an environment by moving objects (Batra et al., 2020).

We see several limitations and areas for future work. First, we examined ground-based navigation agents operating in digitizations of real houses. This limits the agent to a 2D manifold and induces strong structural priors on environment layout. As such, it is unclear how our results generalize to a drone flying through a large forest. Second, we examined agents with a minimal perceptual system. In the supplementary text, we attempted to decode occupancy grids (metric maps) from `Depth` sensor equipped agents and did not find convincing evidence. Our conjecture is that for higher-level strategies like map-building to emerge, the learning problem must not admit 'trivial' solutions. We believe that the minimal perception system used in our work also served to create such a challenging learning problem. Third, our experiments do not study the effects of actuation noise, which is an important consideration in both robot navigation systems and path integration in biological systems. Fourth, we examine an implicit map-building mechanism (an LSTM), a similar set of experiments could be performed for agents with a differentiable read/write map but no direct mapping supervision. Fifth, our agents only explore their environment for a short period of time (an episode) before their memory is reset. Animals and robots at deployment experience their environment for significantly longer periods of time. Finally, we do not provide a complete mechanistic account for how the agent learns to build its map or what else it stores in its memory.

**Reproducibility Statement:** Implementation details of our analyses are provided in the appendix. Our work builds on datasets and code that are already open-sourced. Our analysis code will be open-sourced.

**Acknowledgements:** We thank Abhishek Kadian for his help in implementing the first version of the SecondNav(**T**→**S**) probe experiment. We thank Jitendra Malik for his feedback on the draft and guidance. EW was supported in part by an ARCS fellowship. The Georgia Tech effort was supported in part by NSF, ONR YIP, and ARO PECASE. The Oregon State effort was supported in part by the DARPA Machine Common Sense program. The views and conclusions contained herein are those of the authors and should not be interpreted as necessarily representing the official policies or endorsements, either expressed or implied, of the U.S. Government, or any sponsor.

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

## A    METHODS AND MATERIALS

### A.1    POINTGOAL NAVIGATION TRAINING

**Task.** In PointGoal Navigation, the agent is tasked with navigating to a point specified relative to its initial location, i.e an input of $(\delta x, \delta y)$ corresponds to going $\delta x$ meters forward and $\delta y$ meters to the right. The agent succeeds if it predicts the stop action within 0.2 meters of the specified point. The agent has access to 4 low-level actions – move_forward (0.25 meters), turn_left ($10°$), turn_right ($10°$), and stop. There is no noise in the agent's actuations.

**Sensors.** The agent has access to solely an idealized GPS+Compass sensor that provides it heading and position *relative* to the starting orientation and location at each time step. There is no noise in the agent's sensors.

**Architecture.** The agent is parameterized by a 3-layer LSTM (Hochreiter & Schmidhuber, 1997) with a 512-d hidden dimension. At each time-step, the agent receives observations $g$ (the location of the goal relative to start), GPS (its current position relative to start), and compass (its current heading relative to start). We also explicitly give the agent an indicator of if it is close to goal in the form of $\min(||g - GPS||, 0.5)$ as we find the agent does not learn robust stopping logic otherwise. All 4 inputs are projected to 32-d using separated fully-connected layers. These are then concatenated with a learned 32-d embedding of the previous action taken to form a 160-d input that is then given to the LSTM. The output of the LSTM is then processed by a fully-connected layer to produce a softmax distribution of the action space and an estimate of the value function.

**Training Data.** We construct our training data based on the Gibson (Xia et al., 2018) and Matterport3D dataset (Chang et al., 2017). We training on 411 scenes from Gibson and 72 from Matterport3D.

**Training Procedure.** We train our agents using Proximal Policy Optimization (PPO) (Schulman et al., 2017) with Generalized Advantage Estimation (GAE) (Schulman et al., 2016). We use Decentralized Distributed PPO (DD-PPO) (Wijmans et al., 2020) to train on 16 GPUs. Each GPU/worker collects 256 steps of experience from 16 agents (each in different scenes) and then performs 2 epochs of PPO with 2 mini-batchs per epoch. We use the Adam optimize (Kingma & Ba, 2015) with a learning rate of $2.5 \times 10^{-4}$. We set the discount factor $\gamma$ to 0.99, the PPO clip to 0.2, and the GAE hyper-parameter $\tau$ to 0.95. We train until convergence (around 2 billion steps of experience).

At every timestep, $t$, the agent is in state $s_t$ and takes action $a_t$, and transitions to state $s_{t+}$. It receives shaped reward in the form:

$$r_t = \begin{cases} 2.5 \cdot \text{Success} & \text{if } a_t \text{ is } \texttt{Stop} \\ -\Delta_{\text{geo\_dist}}(s_t, s_{t+1}) - \lambda & \text{Otherwise} \end{cases} \tag{1}$$

where $\Delta_{\text{geo\_dist}}(s_t, s_{t+1})$ is the change in geodesic (shortest path) distance to goal between $s_t$ and $s_{t+1}$ and $\lambda$=0.001 is a slack penalty encouraging shorter episodes.

**Evaluation Procedure.** We evaluate the agent in the 18 scenes from the Matterport3D test set. We use the episodes from Savva *et al.* (Savva et al., 2019), which consist of 56 episodes per scene (1008 in total). Episode range in distance from 1.2 to 30 meters. The ratio of geodesic distance to euclidean distance between start and goal is restricted to be greater than or equal to 1.1, ensuring that episodes are not simple straight lines. Note that reward is *not* available during evaluation.

The agent is evaluated under two metrics, Success, whether or not the agent called the `stop` action with 0.2 meters of the goal and Success weighted by normalized inverse Path Length (SPL) (Anderson et al., 2018). SPL is calculated as follows: given the agent's path $[s_1, \ldots, s_T]$ and the initial geodesic distance to goal $d_i$ for episode $i$, we first compute the length of the agent's path

$$l_i = \sum_{t=2}^{T} ||s_t - s_{t-1}||_2 \tag{2}$$

then SPL for episode $i$ as

$$\text{SPL}_i = \text{Success}_i \cdot \frac{d_i}{\min\{d_i, l_i\}} \tag{3}$$

We then report SPL as the average of $\text{SPL}_i$ across all episodes.

## A.2 PROBE TRAINING

**Task.** The probe task is to either navigate from start to goal again (SecondNav(**S**→**T**)) or navigate from goal to start (SecondNav(**T**→**S**)). For SecondNav(**S**→**T**), the probe is initialized at the starting location but *with* the agent's final heading. For SecondNav(**T**→**S**), the probe is initialized with the agent's final heading and position. In both cases, the probe and the agent share the same coordinate system – *i.e.* in SecondNav(**T**→**S**), the initial GPS and Compass readings for the probe are identical the the final GPS and Compass readings for the agent. When the agent does not successfully reach the goal, the probe task is necessarily undefined and we do not instantiate a probe.

**Sensors, Architecture, Training Procedure, Training Data.** The probe uses the same sensor suite, architecture, training procedure, and training data as the agent, described in Section A.1

Note that no gradients (or rewards) follow from probe to agent. From the agent's perspective, the probe does not exist. From the probe's perspective, the agent provides a dataset of initial locations (or goals) and initial hidden states.

**Evaluation Procedure.** We evaluate the probe in a similar manner the agent except that any episode which the agent is unable to complete (5%) is removed due to the probe task being undefined if the agent is unable to complete the task. The agent reaches the goal 95% of the time, thus only 50 out of 1008 possible probe evaluation episodes are invalidated. The control probe type accounts for this. We ignore the agent's trajectory when computing SPL for the probe.

## A.3 OCCUPANCY MAP DECODING

**Task.** We train a decoding network to predict the top-down occupancy map of the environment from the final internal state of the agent $(\mathbf{h}_t, \mathbf{c}_t)$. We limit the decoder to only predict within 2.5 meters of any location the agent visited.

**Architecture.** The map-decoder is constructed as follows: First the internal state $(\mathbf{h}_t, \mathbf{c}_t)$ is concatenated into a $512 \times 6$-d vector. The vector is then passed to a 2-layer MLP with a hidden dimension of $512$-d that produces a $4608$-d vector. This $4608$-d vector is then reshaped into a $[128, 6, 6]$ feature-map. The feature map is processed by a series of Coordinate Convolution (`CoordConv`) (Liu et al., 2018) Coordinate Up-Convolution (`CoordUpConv`) layers decrease the channel-depth and increase spatial resolution to $[16, 96, 96]$. Specifically, after an initial `CoordConv` with an output channel-depth of 128, we use a series of 4 `CoordUpConv`-`CoordConv` layers where each `CoordUpConv` doubles

the spatial dimensions (quadruples spatial resolution) and each `CoordConv` reduces channel-depth by half. We then use a final 1x1-Convolution to create a $[2, 96, 96]$ tensor representing the non-normalized log-probabilities of whether or not an given location is navigable or not.

Each `CoordConv` has kernel size 3, padding 1, and stride 1. `CoordUpConv` has kernel size 3, padding 0, and stride 2. Before all `CoordConv` and `CoordUpConv`, we use 2D Dropout (Srivastava et al., 2014; Tompson et al., 2015) with a zero-out probability of 0.05. We use Batch Normalization layers (Ioffe & Szegedy, 2015) and the ReLU activation function (Nair & Hinton, 2010) after all layers except the terminal layer.

**Training Data.** We construct our training data by having a trained agent perform episodes of Point-Goal navigation on the training dataset. Note that while evaluation is done utilizing the final hidden state, we construct our training dataset by taking 30 time steps (evenly spaced) from the trajectory and ensuring the final step is included.

**Training Procedure.** We train on 8 GPUs with a batch size of 128 per GPU (total batch size of 1024). We use the AdamW optimizer (Kingma & Ba, 2015; Loshchilov & Hutter, 2019) with an initial learning rate of $10^{-3}$ and linearly scale the learning rate to $1.6 \times 10^{-2}$ over the first 5 epochs (Goyal et al., 2017) and use a weight-decay of $10^{-5}$. We use the validation dataset to perform early-stopping. We use Focal Loss (Lin et al., 2017) (a weighted version of Cross Entropy Loss) with $\gamma = 2.0$, $\alpha_{\text{NotNavigable}} = 0.75$, and $\alpha_{\text{Navigable}} = 0.25$ to handle the class imbalance.

**Evaluation Data and Procedure.** We construct our evaluation data using the validation dataset. Note that the scenes in evaluation are novel to both the agent and the decoder. We evaluate the predicted occupancy map from the final hidden state/final time step. We collect a total of 5,000 episodes.

### A.4 PAST AND FUTURE POSITION PREDICTION

**Task.** We train a decoder to predict the change in agent location given the internal state at time t $(\mathbf{h}_t, \mathbf{c}_t)$. Specifically, let $s_t$ be the agent's position at time $t$ where the coordinate system is defined by the agent's starting location (*i.e.* $s_0 = \mathbf{0}$), and $s_{t+k}$ be its position $k$ steps into the future/past, then the decoder is trained to model $f((\mathbf{h}_t, \mathbf{c}_t)) = s_{t+k} - s_t$.

**Architecture.** The decoder is a 3-layer MLP that produces a 3 dimensional output with hidden sizes of 256 and 128. We use Batch Normalization (Ioffe & Szegedy, 2015) and the ReLU activation function (Nair & Hinton, 2010) after all layers except the last.

**Training Data.** The training data is collected from executing a trained agent on episodes from the training set. For each episode, we collect all possible pairs of $s_t, s_{t+k}$ for a given value of $k$.

**Training Procedure.** We use the AdamW optimizer (Kingma & Ba, 2015; Loshchilov & Hutter, 2019) with a learning rate of $10^{-3}$, a weight decay of $10^{-4}$, and a batch size of 256. We use a Smooth L1 Loss/Huber Loss (Huber, 1964) between the ground-truth change in position and the predicted change in position. We use the validation set to perform early stopping.

**Evaluation Procedure.** We evaluate the trained decoded on held-out scenes. Note that the held-out scenes are novel both to the agent and the decoder.

**Visualization of Predictions.** For visualization the predictions of past vitiation, we found it easier to train a second decoder that predicts *all* locations the agent visited previously on a 2D top down map given the internal state $(\mathbf{h}_t, \mathbf{c}_t)$. This decoder shares the exact same architecture and training procedure as the occupancy grid decoder. The decoder removes the temporal aspect from the prediction, so it is ill-suited for any time-dependent analysis, but produces clearer visualizations.

**Excursion Calibrated Analysis.** To perform the excursions forgetting analysis, we use the excursion labeled episodes. We marked the end of the excursion as the last 10% of the steps that are part of the excursion. For a given point in time $t$, we classify that point into one of {Non-Excursion, Excursion, Exit}. We then examine how well this point is remembered by calculating the error of predicting the point $t$ from $t + k$, *i.e.* how well can $t$ be predicted when it is $k$ steps into the past. When $t$ is part of an excursions (both the excursion and the exit) we limit $t + k$ to either be part of the same excursion or not part of an excursion. When $t$ is not part of an excursion, $t + k$ must also not be part of an excursion nor can there be any excursion in the range $[t, t + k]$.

### A.5 Collision Prediction Linear Probe

**Task.** The task of this probe is to predict of the previous action taken lead to a collision given the current hidden state. Specifically it seeks to learn a function $\text{Collided}_t = f((\mathbf{h}_t, \mathbf{c}_t))$ where $(\mathbf{h}_t, \mathbf{c}_t)$ is the internal state at time $t$ and $\text{Collided}_t$ is whether or not the previous action, $a_{t-1}$ lead to a collision.

**Architecture.** The architecture is logistic classifier that takes the concatentation of the hidden state and produces log probability of $\text{Collided}_t$.

**Training Data.** We construct our training data by having a trained agent perform episodes of Point-Goal navigation on the training set. We collect a total of 10 million samples and then randomly select 1 million for training. We then normalize each dimension independently by computing mean and standard deviation and then subtract mean and divide by standard deviation. This ensures that all dimensions have the same average magnitude.

**Training Procedure.** We training on 1 GPU with a batch size of 256. We use the Adam optimizer (Kingma & Ba, 2015) with a learning rate of $5 \times 10^{-4}$. We train for 20 epochs.

**Evaluation Data and Procedure.** We construct our evaluation data using the same procedure as the training data, but on the validation dataset and collect 200,00 samples (which is then subsampled to 20,000).

**Important Dimension Selection.** To select which dimensions are important for predicting collsions, we re-train our probe with various $\ell_1$ penalties. We sweep from 0 to 1000 and then select the penalty that results in the lowest number of significant dimensions without substantially reducing accuracy. We determine the number of significant dimensions by finding the smallest number of dimensions we can keep while maintaining 99% of the full network's performance. This is achieved by sorting the dimensions by the $\ell_1$ norm of their weight in descending order and adding dimensions (in sorted order) until ¿99% of the full network's performance is achieved.

**The t-SNE manifold** is computed using 20,000 samples. This is then randomly subsampled to 1,500 for visualization. We use the techniques proposed in Kobak & Berens (2019) to set our t-SNE parameters.

### A.6 Data and Materials Availability

The Gibson (Xia et al., 2018) and Matterport3D (Chang et al., 2017) datasets can be acquired from their respective distributors. Habitat (Savva et al., 2019) is open source. The environment in the image in Fig. 3 is from the Replica Dataset (Straub et al., 2019).

## B Additional Discussions

### B.1 Relationship to Cognitive Maps

Throughout the text, we use the term 'map' to mean a spatial representation that supports intelligent behaviors like taking shortcuts. Whether or not this term is distinct from the specific concept of a 'cognitive map' is debated.

Cognitive maps, as defined by O'keefe & Nadel (1978), imply a set of properties and are generally attached to a specific mechanism. The existence of a cognitive map requires that the agent be able to reach a desired goal in the environment from any starting location without being given that starting location, *i.e.* be able to navigate *against* a map. Further, cognitive maps refer to a specific mechanism – place cells and grid cells being present in the hippocampus. Other works have also studied 'cognitive maps' and not put such restrictions on its definition (Gallistel, 1990; Tolman, 1948), however these broader definitions have been debated (Jacobs, 2003).

Our work shows that the spatial information contained within the agent's hidden state enables map-like properties – a secondary agent to take shortcuts through previously unexplored free space – and supports the decoding of a metric map. However, these do not fully cover the proprieties of O'keefe & Nadel (1978)'s definition nor do we make a mechanistic claim about how this information is stored in the neural network, though we do find the emergence of collision-detection neurons.

## C  ADDITIONAL EXPERIMENTS

### C.1  BLIND SHORTEST PATH NAVIGATION WITH TRUE STATE

In the main text, we posited that blind agents learn wall-following as this an effective strategy for blind navigation in *unknown* environments. We posit that this is because the agent does not have access to true state (it does not know the current environment nor where it is in global coordinates). In this experiment we show that blind agents learn to take shortest paths, as opposed to wall-following, when trained in a single environment (implicitly informing the agent of the current environment) and uses the global coordinate system. [8]

We use an identical agent architecture and training procedure as outline for PointGoal navigation training in the Materials and Methods with two differences: 1) A single training and test environment and 2) usage of the global coordinates within the environment for both goal specific and the agent's GPS+Compass sensor. We perform this experiment on 3 scenes, 1 from the Gibson val dataset and 2 from Matterport3D val dataset. The average SPL during training is $99_{\pm 0.1}$ showing that the blind agent learns shortest path navigation not wall-following. Figure A6 shows examples of an agent trained in a single scene with global coordinates and an agent trained in many scenes with episodic coordinates.

These two settings, i) where the agent uses an episodic coordinate system and navigates in unknown environments, and ii) where the agent uses global coordinates and navigates in a known environment can be seen as the difference between a partially observable Markov decision process (POMDP) and a Markov decision process (MDP). In the POMDP case, the agent must learn a generalizable policy while it can overfit in the MDP case.

### C.2  FURTHER ANALYSIS OF THE PROBE'S PERFORMANCE

In the main text, we showed that the probe is indeed much more efficient than the agent, but how is this gain achieved? Our hypothesis is that the probe improves upon the agent's path by taking shortcuts and eliminating excursions (representing an 'out and back'). We define an excursion as a sub-path that approximately forms a loop. To quantify excursions, we manually annotate excursions in 216 randomly sampled episodes in evaluation environments. Of the labeled episodes, 62% have a least 1 excursion. On average, an episode has 0.95 excursions, and excursions have an average length of 101 steps (corresponding to 8.23 meters). Since excursions represent unnecessary portions of the trajectory, this indicates that the probe should be able improve upon the agent's path by removing these excursions.

We quantify this excursion removal via the normalized Chamfer distance between the agent's path and the probe's path. Formally, given the agent's path $\mathtt{Agent}=[s_1^{(\text{agent})}, \ldots, s_T^{(\text{agent})}]$ and the probe's path $\mathtt{Probe}=[s_1^{(\text{probe})}, \ldots, s_N^{(\text{probe})}]$ where $s \in R^3$ is a point in the environment:

$$\text{PathDiff}(\mathtt{Agent}, \mathtt{Probe}) = \frac{1}{N} \sum_{i=1}^{N} \min_{1 \leq j \leq T} \text{GeoDist}(s_i^{(\text{agent})}, s_j^{(\text{probe})}), \tag{4}$$

where $\text{GeoDist}(\cdot, \cdot)$ indicates the geodesic distance (shortest traverseable path-length).

Note that Chamfer distance is not symmetric. PathDiff(Probe, Agent) measures the average distance of a point on the probe path $s_j^{(\text{probe})}$ from the closest point on the agent path. A large PathDiff(Probe, Agent) indicates that the probe travels through novel parts of the environments (compared to the agent). Conversely, PathDiff(Agent, Probe) measures the average distance of a point on the agent path $s_i^{(\text{agent})}$ from the closest point on the probe path. A large $\big(\text{PathDiff}(\mathtt{Agent}, \mathtt{Probe}) - \text{PathDiff}(\mathtt{Probe}, \mathtt{Agent})\big)$ gap indicates that agent path contains excursions while the probe does not; thus, we refer to this gap as Excursion Removal. To visually understand why this is the case, consider the example agent and probe paths in Fig. A7. Point (C) lies on an excursion in the agent path. It contributes a term to PathDiff(Agent, Probe) but not to PathDiff(Probe, Agent) because (D) is closer to (E) than (C).

---

[8]Recall that in the episodic coordinate system the origin is defined by the agent's starting position and orientation. In the global coordinate system the origin is an arbitrary but consistent location (we simply use the origin for a given scene defined in the dataset). Thus in the global coordinate system the goal is specified as *'Go to $(x, y)$'* where $x$ and $y$ are specified in the global coordinate system, not with respect to the agent's current location.

On both SecondNav(**S**→**T**) and SecondNav(**T**→**S**), we find that as the efficiency of a probe increases, Excursion Removal also increases (Table A2, row 1 vs. 2, 2 vs. 3), confirming that the `TrainedAgentMemory` probe is more efficient *because* it removes excursions.

We next consider if the `TrainedAgentMemory` probe also travels through previously unexplored space in addition to removing excursions. To quantify this, we report PathDiff(`Probe`, `Agent`) on episodes where agent SPL is less than average (less than 62.9%).[9] If probes take the same path as the agent, we would expect this metric to be zero. If, however, probes travel through previously unexplored space to minimize travel distance, we would expect this metric to be significantly non-zero. Indeed, on SecondNav(**S**→**T**), we find the `TrainedAgentMemory` probe is 0.32 meters away on average from the closest point on the agent's path (99% empirical bootstrap of the mean gives a range of $(0.299, 0.341)$). See Fig. A7 for a visual example. On SecondNav(**T**→**S**), this effect is slightly more pronounced, the `TrainedAgentMemory` probe is 0.55 meters away on average (99% empirical bootstrap of the mean gives a range of $(0.52, 0.588)$). Taken holistically, these results show that the probe is both more efficient than the agent *and* consistently travels through new parts of the environment (that the agent did not travel through). Thus, the spatial representation in the agent's memory is not simply a 'literal' episodic summarization, but also contains anticipatory inferences about previously unexplored spaces being navigable (*e.g.* traveling along the hypotenuses instead of sides of a room).

In the text above we reported free space inference only on episodes where the agent gets an SPL bellow average. In Fig. A12 we provide a plot of Free Space Inference vs. Agent SPL to show the impact of other cutoff points. In Fig. A13 we also provide a similar plot of Excursion Removal vs. Agent SPL. In both cases, as agent SPL increase, the probe is able to infer less free space or remove less excursions.

### C.3 FUTURE VISITATION PREDICTION

In the main text we examined what types of systematic errors are made when decoding past agent locations, here we provide addition analysis and look at predicting future observations as that will reveal if there are any idiosyncrasies in what can be predicted about future *vs*. what will happen in the future.

Given ground truth location $s_{t+k}$, we evaluate the decoder via i) absolute L2 error $||\hat{s}_{t+k} - s_{t+k}||$ and ii) relative L2 error $||\hat{s}_{t+k} - s_{t+k}||/||s_{t+k} - s_t||$. To determine baseline (or chance) performance, we train a second set of decoders where instead of using the correct internal state $(\mathbf{h}_t, \mathbf{c}_t)$ as the input, we randomly select an internal state from a *different* trajectory. This will evaluate if there are any inherent biases in the task.

In Fig. A8, we find that the decoder is able to accurately predict where the agent has been, even for long time horizons – *e.g.* at 100 time steps in the past, relative error is 0.55 and absolute error is 1.0m, compared to relative error of 1.0 and absolute error of 3.2m for the chance baseline prediction. For short time horizons the decoder is also able to accurately predict where the agent will be in the future – *e.g.* at 10 time steps into the future, relative and absolute error are below chance. Interestingly, we see that for longer range future predictions, the decoder is worse than chance in relative error but on-par in absolute error. This apparent contradiction arises due to the decoders making (relatively) large systematic errors when the agent backtracks. In order for the decoder to predict backtracking, the agent would need to already know its future trajectory will be sub-optimal (*i.e.* lead to backtracking) but still take that trajectory. This is in contradiction with the objective the agent is trained for, to reach the goal as quickly as possible, and thus the agent would not take a given path if it knew it would lead to backtracking.

### C.4 EXTENSION TO SIGHTED NAVIGATION AGENTS

In the main text we analyzed how 'blind' agents, those with limited perceptual systems, utilize their memory and found evidence that they build cognitive maps. Here, we extend our analysis to agents with rich perceptual systems, those equipped with a `Depth` camera and an egomotion sensor. Our primary experimental paradigm relies on showing that a probe is able to take shortcuts when given the agent's memory. This experimental paradigm relies on the probe being able to take a shorter

---

[9]We restrict to a subset where the agent has relatively low SPL to improve dynamic range. When the agent has high SPL, there won't be excursions to remove and this metric will naturally be low. In the supplementary text we provide plots of this metric vs. agent SPL.

path than the agent. Navigation agents with vision can perform PointNav near-perfectly (Wijmans et al., 2020) and thus there isn't room for improving, rendering this experiment infeasible. As a supplement to this experiment, we also show that a metric map (top-down occupancy grid) can be decoded from the agents memory. This procedure can also be applied to sighted agents.

We use the ResNet50 (He et al., 2016) Gibson-2plus (Xia et al., 2018) pre-train model from Wijmans *et al.* (Wijmans et al., 2020) and train an occupancy grid decoder using the same procedure as in the main text. Note however we utilize only Gibson for training and the Gibson validation scenes as held-out data instead of Matterport3D as this agent was only trained on Gibson. As before, we compare performance from `TrainedAgentMemory` with `UntrainedAgentMemory`.

We find mixed results. When measuring performance with Intersection-over-Union (IoU), `UntrainedAgentMemory` *outperforms* `TrainedAgentMemory` (40.1% vs. 42.9%). However, when measuring performance with average class balanced accuracy, `TrainedAgentMemory` outperforms `UntrainedAgentMemory` (61.8% vs. 53.1%). Fig. A9 and Fig. A10 show the corresponding distribution plots.

Overall, this experiment does not provide convincing evidence either way to whether vision-equipped agents build metric maps in their memory. However, it does show that vision-equipped agents, if they do maintain a map of their environment, create one that is considerably more challenging to decode. Further, we note this does not necessarily imply similarly mixed results as to whether or not vision agents maintain a still spatial but sparser representation, such as a topological graph, as their rich perception can fill in the details in the moment.

## C.5   NAVIGATION FROM MEMORY ALONE

In the main text we showed that agents learn to build map-like representations. A map-like representation of the environment, should, to a degree, support navigation with no external information, *i.e.* by dead reckoning. Given that the actions are deterministic, the probe should be able to perform either task without external inputs and only the agent's internal representation and the previously taken action. The localization performed by the probe in this setting is similar to path integration, however, it must also be able to handle any collisions that occur when navigating.

Fig. A11 shows performance vs. episode length for SecondNav($\mathbf{S} \rightarrow \mathbf{T}$) and SecondNav($\mathbf{T} \rightarrow \mathbf{S}$). There are two primary trends. For short navigation episodes ($\leq$5m), the agent is able to complete the task often. We also find that under this setting, SecondNav($\mathbf{T} \rightarrow \mathbf{S}$) is an easier task. This is due to the information conveyed to the probe by its initial heading. In SecondNav($\mathbf{T} \rightarrow \mathbf{S}$), the probe can make progress by simply turning around and going forward, while in SecondNav($\mathbf{S} \rightarrow \mathbf{T}$), the final heading of the agent is not informative of which way the probe should navigate initially. Overall, these results show that the representation built by the agent is sufficient to navigate short distances with *no external information*.

**Experiment procedure.** This experiment mirrors the probe experiment described in methods and materials with three differences: 1) The input from the GPS+Compass sensor is zero-ed out. 2) The change in distance to goal shaping in the reward is normalized by the distance from initial state to goal. We find that the prediction of the value function suffers considerably otherwise. 3) An additional reward signal as to whether or not the last action taken decreased the angle between the probe's current heading and the direction along the shortest path to goal is added. We find the probe has challenges learning to turn around on the SecondNav($\mathbf{T} \rightarrow \mathbf{S}$) task otherwise (as it almost always starts facing $180°$ in the wrong direction).

Let $h_t^{gt}$ be the heading along the shortest path to goal from the probe's current position $s_t$, $h_t$ be the probe's current heading, then $\text{AngularDistance}(h_t^{gt}, h_t)$ is the error in the probe's heading. The full reward for this probe is then

$$r_t(s_t, a_t, s_{t+1}) = \begin{cases} 2.5 \cdot \text{Success} & \text{if } a_t \text{ is Stop} \\ -10.0 \cdot \Delta_{\text{geo\_dist}}(s_t, s_{t+1})/\text{GeoDist}(s_0, g) \\ \quad -0.25 \cdot \Delta_{\text{HeadingError}}(s_t, s_{t+1}) \\ -\lambda & \text{Otherwise} \end{cases} \quad (5)$$

### C.6 MEMORY LENGTH

The method presented in the main text to examine memory length is post-hoc analysis performed on the 'blind' PointGoal Navigation agents and thus the agent is operating out-of-distribution. From the agent's view, it is still performing a valid PointGoal navigation episode, just with a different starting location, but the agent may not have taken the same sequence of actions if started from that location. While we would still expect performance to stature with a small $k$ if the memory length is indeed short, it is imprecise with measuring the exact memory length of the agent and does not answer what memory budget is required to perform the task.

Here we examined *training* agents with a fixed memory length LSTM. Fig. A14 shows similar trends to those described in the main paper – performance increases as the memory budget increases – however performance is higher when the agent is trained for a given memory budget. Due to the increased compute needed to train the model (*e.g.* training a model with a memory length of 128 is $128\times$ more computationally costly), we where unable to train for a memory budget longer than 256.

We also note the non-monotonicity in Fig. A14. We conjecture that this is a consequence of inducing the negative effects of large-batch optimization (Keskar et al., 2017) – training with a memory budget of $k$ effectively increases the batch size by a factor of $k$. Keeping the batch size constant has its own drawbacks; reducing the number of parallel environments will harm data diversity and result in overfitting while reducing the rollout length increases the bias of the return estimate and makes credit assignment harder. Thus we kept number of environments and rollout length constant.

## D SUPPLEMENTARY VIDEOS

**Movies S1-3** Videos showing blind agent navigation with the location of the hidden state in the collision t-SNE space. Notice that the hidden state stays within a cluster throughout a series of actions.

|   | Probe Type | SecondNav(**S**→**T**) Excursion Removal | SecondNav(**T**→**S**) Excursion Removal |
|---|---|---|---|
| 1 | `AllZeroMemory` | $0.21_{\pm0.017}$ | $0.21_{\pm0.004}$ |
| 2 | `UntrainedAgentMemory` | $0.23_{\pm0.009}$ | $0.25_{\pm0.009}$ |
| 3 | `TrainedAgentMemory` | $0.52_{\pm0.014}$ | $0.51_{\pm0.011}$ |

Table A2: **Excursion removal** result of our trained probe agent under three configurations – initialized with an empty representation (`AllZeroMemory`), a representation of a random agent walked along the trained agent's path (`UntrainedAgentMemory`), and the final representation of the trained agent (`TrainedAgentMemory`). 95% confidence interval reported over 5 agent-probe pairs.

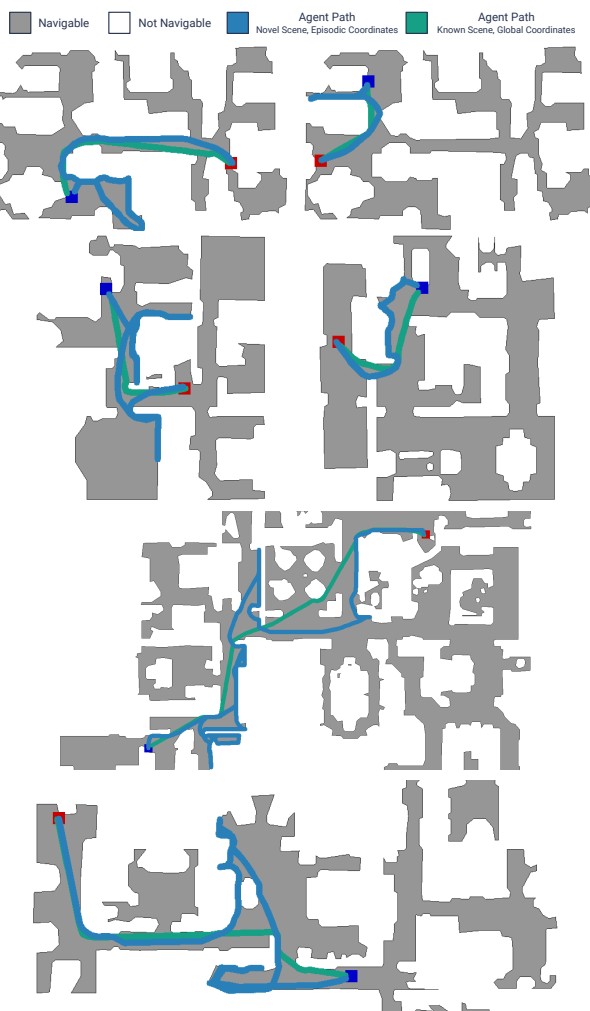

Figure A6: **True state trajectory comparison.** Example trajectories of an agent with true state (trained for a specific environment and using global coordinates), green line, compared to an agent trained for many environments and using episodic coordinates, blue line. The later is what we examine in this work. Notice that the agent with true state take shortest path trajectories while the agent without true state instead exhibits strong wall-following behavior.

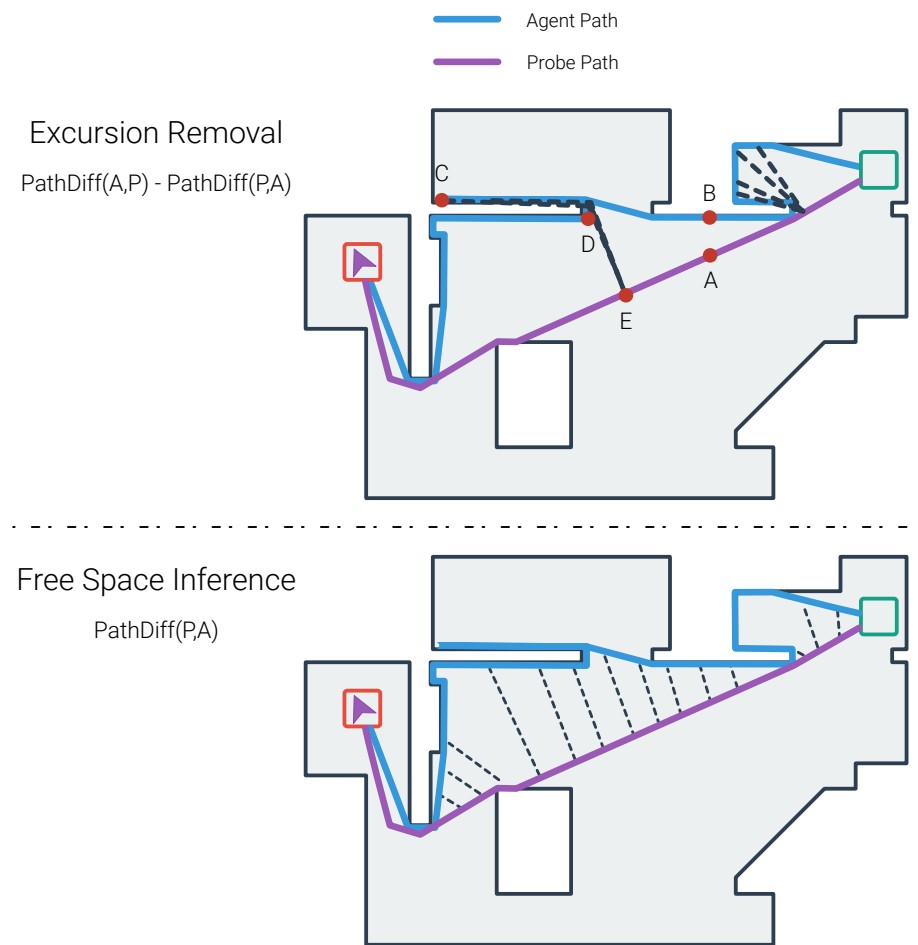

Figure A7: **Two categories of probe shortcut.** *'Excursion Removal'* is when the probe removes excursions from the agent's path. The dashed line shows the distance between the points in the excursion and the closest point in the probe's path. *'Free Space Inference'* occurs when the probe travels through previously unvisited locations in the environments. The dashed lines show the distance between any points in the probe's path and the closest point in the agent's path.

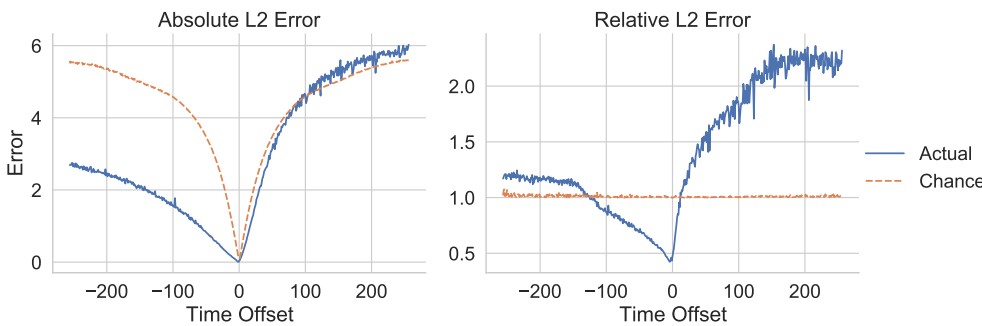

Figure A8: **Past and future prediction.** Performance of decoders trained to predict where the agent was in the past/will be in the future. On the x-axis is how far into the past or future the decoder is predicting (positive values are future predictions and negative values are past predictions). The y-axis is either absolute or relative L2 error between the predicted location of the agent and the true location.

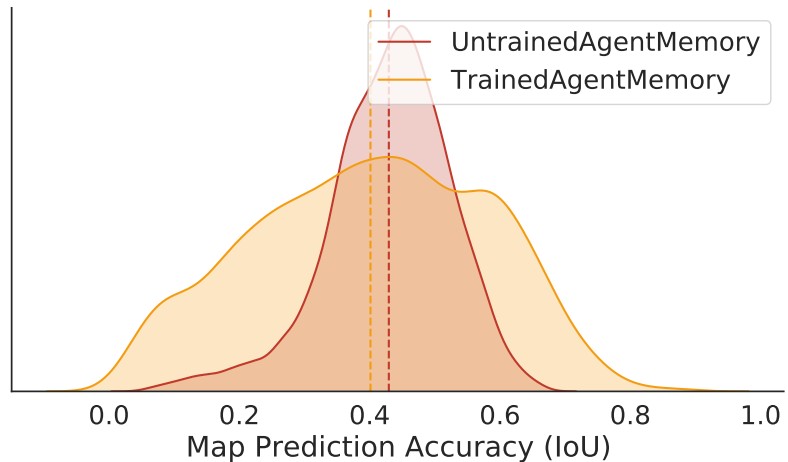

Figure A9: Map prediction accuracy (Intersection over Union) for `Depth` sensor equipped agents.

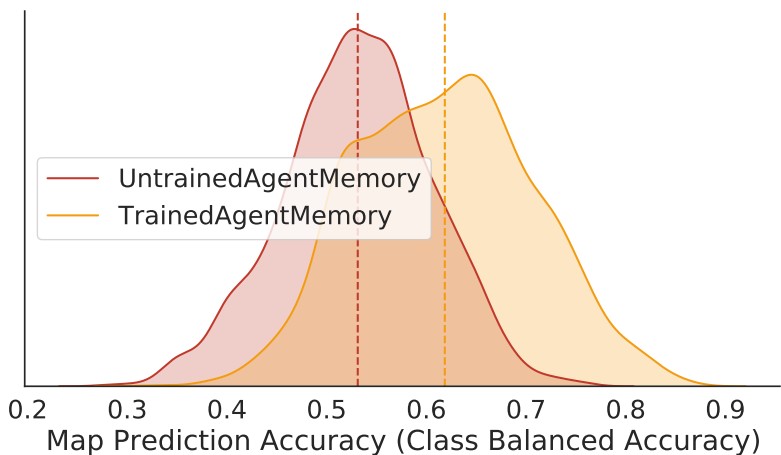

Figure A10: Map prediction accuracy (class balanced accuracy) for `Depth` sensor equipped agents.

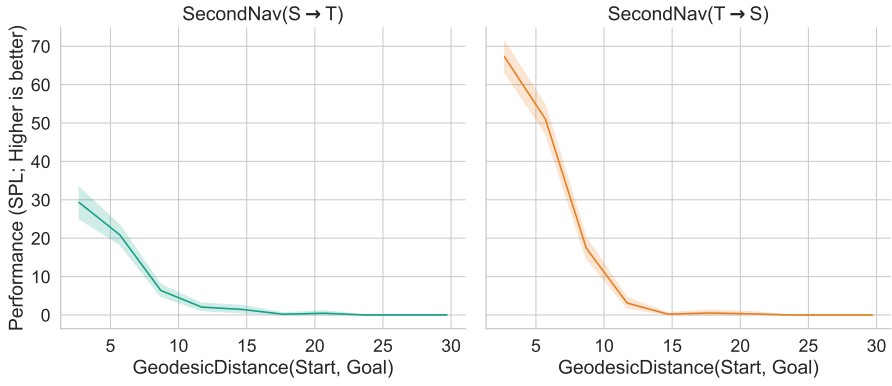

Figure A11: **Memory-only probe performance.** Performance (in SPL; higher is better) as a function of geodesic distance from start to goal for the `TrainedAgentMemory` probe without inputs on SecondNav(**S**→**T**) and SecondNav(**T**→**S**). More information can be found under the 'Navigation from memory alone' header.

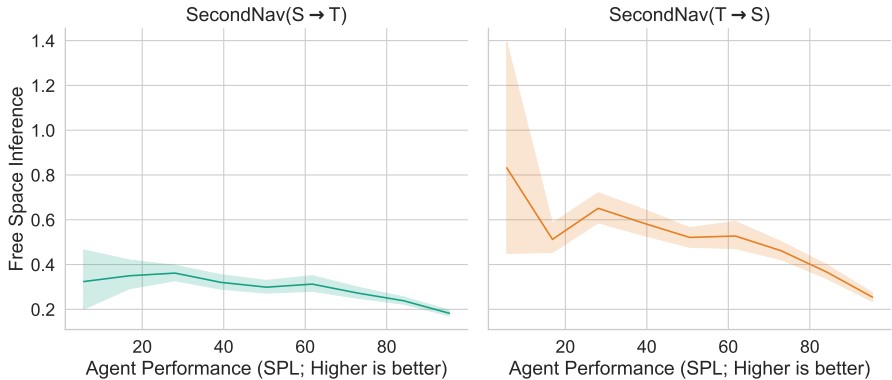

Figure A12: Free Space Inference for the `TrainedAgentMemory` probe on both SecondNav(**S**→**T**) and SecondNav(**T**→**S**) as a function of agent SPL. We see that as agent SPL decreases, the probe is able to take paths that inference more free space.

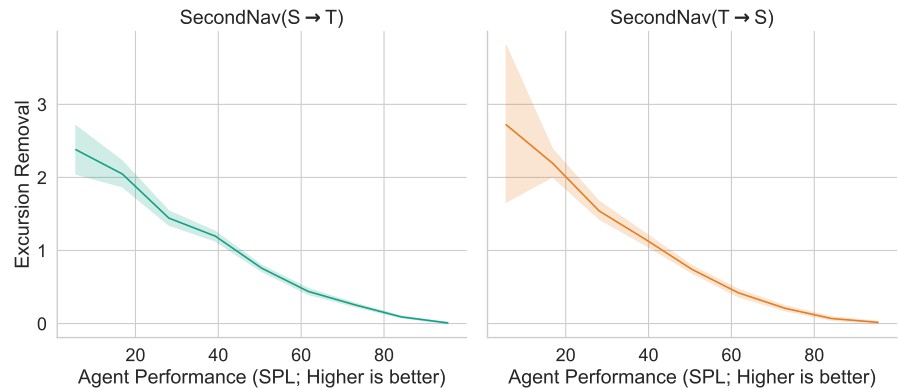

Figure A13: Excursion Removal for the `TrainedAgentMemory` probe on both SecondNav(**S**→**T**) and SecondNav(**T**→**S**) as a function of agent SPL. We see that as agent SPL decreases, excursion removal increases since the probe is able to remove additional excursions.

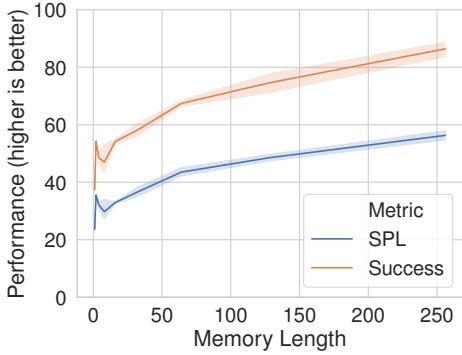

Figure A14: Performance vs. memory length for agents *trained* under a given memory length. Note that longer memory lengths are challenging to train for under this methodology as it induces the negative effects of large-batch optimization and is computationally expensive.

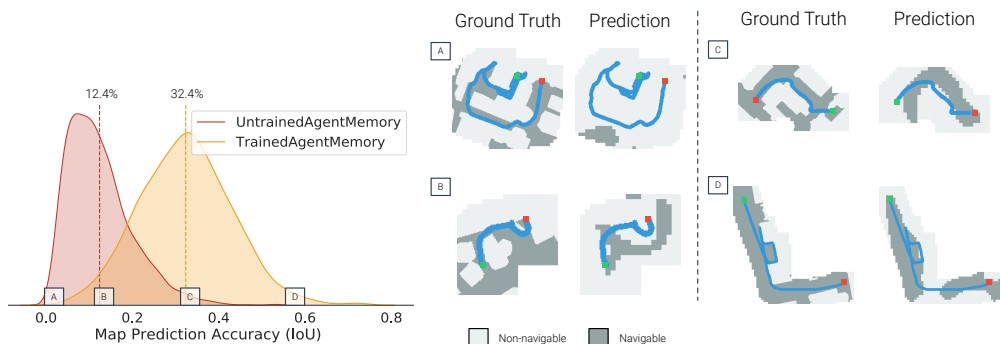

Figure A15: **Map prediction with poor examples**. In the main text we shows qualitative examples for the average prediction and a good prediction. Here we show two additional examples: A, a very poor quality prediction. This shows that the decoder sometimes does make large mistakes. B, the average prediction for the UntrainedAgentMemory decoder. This shows the qualitative difference between the average UntrainedAgentMemory and TrainedAgentMemory prediction.

