# OpenReview forum: "Emergence of Maps in the Memories of Blind Navigation Agents"
_ICLR.cc/2023/Conference — ICLR 2023 notable top 25%_

### Official Review · Reviewer_ccRY · 2022-10-13

**Confidence:** 4
**Correctness:** 4
**Technical Novelty And Significance:** 2
**Empirical Novelty And Significance:** 4
**Recommendation:** 8

**Clarity, Quality, Novelty And Reproducibility:**

The results seem like they should be fairly easy to reproduce, generally using existing datasets and packages that have been tested many times before. I appreciate the commitment to releasing the code.

As mentioned above, the work is very easy to follow along and understand and the hypothesis that this paper attempted to solve was very creative, novel, and enjoyable. There isn't a ton of work focused on similar emergent behaviors of embodied agents.

**Strength And Weaknesses:**

Strengths:
- The paper is very well written and structured. It was a pleasure to read. I also appreciate the time that went into the figures and outline.
- The experiments were well thought out and convincing. Both the probing experiments and occupancy map prediction experiments were very interesting and are convincing of the phenomenon of blind agents encoding maps.
- It is surprising and impressive that blind agents are able to achieve 95% success rate on PointNav.
- The question posed in this work is intellectually interesting and creative.
- The experiments controlling for memory length are also great.

Weaknesses:
- **W1.** This paper presents a neat and interesting observation, and a set of experiments to test their hypothesis. However, the implications of this work don't seem very widespread or practically useful for future work. Suppose the hypothesis that model-free agents build maps without explicitly being trained to build them is true. Who cares? Will this change how you train embodied agents in the future?
- **W2.** It's disappointing that the experiments with vision-based agents didn't work out. The hypothesis makes sense, but effectively, the experiments in this paper are on grid-world environments. It would have been nice to show the hypothesis holds with vision-based agents or in more complex tasks like rearrangement.

**Summary Of The Paper:**

The authors test the hypothesis of if memory-free navigation-based agents build maps of their environment. They use blind agents trained to perform PointNav, which are effectively operating in grid-world environments. They show very convincing and creative experiments controlling for the memory length of the agent, predicting occupancy maps, and probing the agent.

**Summary Of The Review:**

I found the paper very enjoyable and really interesting to read. The work isn't going to be very practically useful and won't change much from a practitioner's perspective, but tests a creative hypothesis of if maps emerge from blind navigation-based agents, which appears to be true. Overall, I found the experiments were very well thought out and convincing, and think the paper is a clear accept.

---

> ### Author Response · Authors · 2022-11-15
> **Response to ccRY**
>
> We thank the reviewer for their review. We are glad they found our paper “well written and structured”, a “pleasure to read”, and that they appreciated “the time that went into the figures and outline.” That our experiments are “well thought out and convincing” and “the question posed in this work is intellectually interesting and creative.”
>
> > However, the implications of this work don't seem very widespread or practically useful for future work. Suppose the hypothesis that model-free agents build maps without explicitly being trained to build them is true. Who cares? Will this change how you train embodied agents in the future?
>
> We respectfully disagree. As an example of the widespread interest in the mechanisms underlying spatial navigation, John O’Keefe and the Mosers were awarded a Nobel Prize for their work on cognitive maps in animals and their discovery of cells that constitute a positioning system in the brain. Our results point towards mapping being a natural solution to navigation. This is relevant to a debate in the cognitive science, ethology, robotics, and AI communities started in (at least) 1948. Furthermore, our demonstration of the surprising efficacy of blind agents despite their limitations will serve to contextualize and provide a strong baseline for the further development of embodied agents by helping to clarify the source of agents’ success. Future work in AI can leverage our results in two ways. 1) By incorporating mapping-related inductive biases (architectural components, mapping supervision as an auxiliary reward, etc), or 2) using implicit maps to perform implicit planning.
>
> At a higher level, our results speak to the efficacy of the research program of embodied AI. By training agents in immersive interactive environments at a large-scale, intelligence (in the form of mapping) simply emerges. This is precisely what Smith and Gasser’s embodiment hypothesis states should happen. This provides confidence in continuing to pursue this program for more challenging tasks, environments, and settings.
>
> Please let us know if you have any additional questions.

---

> > ### Comment · Reviewer_ccRY · 2022-11-16
> > **Response to authors**
> >
> > Thanks for providing additional insight. Perhaps the work will help with cognitive science debates or help motivate work at the intersection of cognitive science and embodiment. Practically speaking, however, I still do not find the results of this work that widespread.
> >
> > > Future work in AI can leverage our results in two ways. 1) By incorporating mapping-related inductive biases (architectural components, mapping supervision as an auxiliary reward, etc), or 2) using implicit maps to perform implicit planning.
> >
> > Tons of research has already gone into mapping for navigation with SLAM and more modern approaches. This has been worked on in robotics for decades. Claiming that future work can / should now start to use mapping as an inductive bias because of the results of this paper is a very overreaching and concerning response to me.

---

> > > ### Author Response · Authors · 2022-11-18
> > > **Response to ccRY**
> > >
> > > > Tons of research has already gone into mapping for navigation with SLAM and more modern approaches. This has been worked on in robotics for decades. Claiming that future work can / should now start to use mapping as an inductive bias because of the results of this paper is a very overreaching and concerning response to me.
> > >
> > > We see your high-level point. We agree that SLAM and structure-from-motion has a long history in robotics and vision (as we say in the paper, paragraph 2, first sentence). We certainly aren't trying to claim any role in that or its ongoing development. All we were trying to express is that perhaps our work will contribute to an increased emphasis on directions such as mapping-related inductive biases and combining implicit mapping with implicit planning. We believe our manuscript strikes a fairly balanced tone, but we do not want to over-claim so we are happy to edit any text that the reviewer may have suggestions on.

---

### Official Review · Reviewer_XPSp · 2022-10-24

**Confidence:** 4
**Correctness:** 4
**Technical Novelty And Significance:** 3
**Empirical Novelty And Significance:** 3
**Recommendation:** 8

**Clarity, Quality, Novelty And Reproducibility:**

Claims are clearly stated in the paper and well supported empirically. Experiments are detailed, allowing reproducibility. Previous work has already considered the emergence of mechanisms close to place cells in recurrent agents (Banino et al., Vector-based navigation using grid-like representations in artificial agents), but this work can still be considered as original from the direction of studying blind agents and the diverse experimental studies.

**Strength And Weaknesses:**

Strengths:
1. Understanding the internal representation of scenes from neural agents is an important direction, both to make them more interpretable and to study the strategies that can be purely learnt from data.
2. The context of the work is well defined, and the experiments are detailed and thorough.
3. This paper presents many surprising findings (stated in paper summary) that are important for the Embodied AI community.

Weaknesses:
1. I am not convinced with the point made by authors in Section 5 about the ability to properly decode the occupancy maps even in unseen locations. This might suggest the decoding model rather learns some patterns about the average structure of occupancy maps. Results on this decoding task are still interesting, but maybe less strong. It would be interesting to quantify how biased towards some redundant structure in maps the decoding model is.
2. It would be interesting to consider a comparison with a baseline that shares the same architecture as the blind agent, but composed of an additional explicit map with differentiable write and read operations. This could allow to compare the content of the maps with the ones decoded from the recurrent agent’s hidden state, as well as task performance.

**Summary Of The Paper:**

This paper studies the implicit emergence of mapping in simple recurrent blind agents. Removing the visual modality allows authors to reduce the possible strategies to reach the specified goal and to thus isolate the construction of an internal map. As mentioned in the paper, no technical novelty is proposed, but a study about an important and interesting question, i.e. whether simple recurrent agents can learn to implicitly map scenes, is studied.

The work presents many surprising findings, such as the strong performance of blind agents, the emergence of neurons specialised in collision detection, the impressive long horizon memory developed these agents, the quality of spatial maps decoded from the agents’ hidden state, and finally the capacity to forget information that is not useful for the task at hand. Authors also show that if initialising the hidden state of an agent with the one from another agent that has finished an episode, the new episode navigates more efficiently between the starting and end points, in both directions.

**Summary Of The Review:**

Such empirical study is of high importance for the community. This paper provides many interesting and well supported insights. Allowing to better understand the underlying reasoning of simple recurrent agents can help develop better navigating agents in the future. I thus tend to recommend acceptance, but am still looking for answers from the authors to my remarks

---

> ### Author Response · Authors · 2022-11-15
> **Response to XPSp**
>
> We thank the reviewer for their review. We are glad they found that our paper “presents many surprising findings ... that are important for the Embodied AI community”, that “understanding the internal representation of scenes from neural agents is an important direction”, and our experiments to be “detailed and thorough.”
>
> >  I am not convinced with the point made by authors in Section 5 about the ability to properly decode the occupancy maps even in unseen locations. This might suggest the decoding model rather learns some patterns about the average structure of occupancy maps.
>
> We agree that there is structure in the environments relevant to decoding occupancy grids -- e.g. since many environments feature hallways, the presence of one wall will increase the probability of there being a matching parallel wall. Further, the decoder must use this structure. It is tasked with predicting unseen locations and, without a prior, this is an impossible task.
>
> However, we disagree that average structure is enough. To control for this possibility, we presented the controlled comparison -- (TrainedAgentMemory, decoder) vs. (UntrainedAgentMemory, decoder) -- with identical train/val sets and decoder architecture (the only change is the input memory). If average structure was enough and agent memory wasn’t necessary to predict unseen locations, we’d expect both to perform similarly. However, we find that (TrainedAgentMemory, decoder) achieves 32.4% IoU while (UntrainedAgentMemory, decoder) achieves 12.4% IoU (Fig. 4). This demonstrates that agent memory is necessary to predict unseen locations. In other words, the decoder is not predicting average structure, but uses the combination of structure and trained agent memory.
>
> In summary, we show that trained agent memory is necessary for predicting unseen locations but we do not show that it is sufficient. We have updated the paper to clarify this nuance.
>
> >  It would be interesting to quantify how biased towards some redundant structure in maps the decoding model is.
>
> Interesting idea! We are unaware of a way to programmatically quantify the amount of redundant structure, but are happy to try something if the reviewer has suggestions.
>
> We have included 100 extra examples of decoder outputs in the supplementary materials in the extra-occupancy-decoder-examples folder, ground-truth is on the left and the prediction is on the right.
>
> > It would be interesting to consider a comparison with a baseline that shares the same architecture as the blind agent, but composed of an additional explicit map with differentiable write and read operations. This could allow to compare the content of the maps with the ones decoded from the recurrent agent’s hidden state, as well as task performance.
>
> Interesting idea! While outside the scope of this work and an entire research project in its own scope, such a model could in future be used to answer questions like “If the agent is given a map, what does it choose to record” and allow us to compare implicit vs. explicit maps. Observing what an agent with an explicit map records may also give further insight into what the implicit-map agent records by giving other things to examine. We will add this discussion to the paper.
>
> Please let us know if you have any additional questions.

---

### Official Review · Reviewer_JNho · 2022-10-24

**Confidence:** 5
**Correctness:** 3
**Technical Novelty And Significance:** 4
**Empirical Novelty And Significance:** 3
**Recommendation:** 10

**Clarity, Quality, Novelty And Reproducibility:**

[clarity, quality, and novelty] The paper is easy to read and provides a novel and comprehensive study of the effect(s) in question.

[reproducibility] The paper is mostly reproducible, relying on open source components for most (if not all) the navigation agent training. The authors have said they will provide the remaining analysis code in the event of publication.

**Strength And Weaknesses:**

The paper is incredibly well written and detailed and the experiments are thought provoking. The paper explores an interesting topic that should be interested to anyone---including both AI/ML scientists and also perhaps cognitive scientists---working in the space of navigation. Each experiment effectively answers its respective "question" and is accompanied with convincing visuals. While the paper does not introduce new "techniques" for navigation or training navigation agents, this should not be seen as a limitation of the work and the investigation and the analyses are valuable on their own. I have only a few comments and questions.

I have some potential concerns about the results from Sec. 3 (Memory is used over long horizons). The agent is trained in such a way that the LSTM is expecting to receive all the previous observations from the start. As such, are the experiments providing out-of-distribution signals to the agent? Can the authors comment on the possibility of poor performance being a result of this shift in the nature of the data. Similarly, if the agent were retrained with a time-limited memory horizon (e.g., 100 or 500 steps), would we expect an increase in performance; new experiments are likely not necessary, though clarification on this point will be helpful.

Relatedly, it would be helpful to have a separate curve (perhaps on the same plot) showing the success rate versus memory length in Fig. 2, as I believe this to be a key detail for understanding the performance with limited memory.

Smaller questions and comments
- The line in Sec. 2 "but also that collisions-vs-not are linearly separable in this space": I do not believe that this claim logically follows from the previous clause. It may be true, but seems disconnected.
- The authors may also want to reference "Learning Deployable Navigation Policies at Kilometer Scale from a Single Traversal" by Bruce, et al 2018. The appendix of this paper tries to extract localization information from an agent's latent state, and may also be good to mention in the related work [I also believe another early paper in the VisDoom environment used a similar approach, but I am unable to find it.]
- Introduction: I understand why it is included in the introduction but that "memory" is essential for strong performance is a fairly obvious conclusion even if the results in the respective section are more detailed. Consider rephrasing this question and answer to reflect the nuance of the effect under scrutiny.

**Summary Of The Paper:**

This paper is a study of blind point goal navigation agents---agents provided only a GPS+compass signal and expected to reach an unseen point goal---and their ability to form implicit maps and an understanding of the environment through navigation. The paper is a thoroughly interesting investigation and includes experiments to understand (1) how well blind memory-full agents can navigate despite their significant limitations, (2) how memory can be transplanted to imbue another agent with the ability to navigate more effectively (using that experience), and (3) to decode the agents' memories into occupancy grids as a means of scrutinizing their implicit maps of their surroundings.


**Summary Of The Review:**

Overall, this is an excellent paper. I have only limited changes and suggestions for improvement. The investigation is detailed and thought provoking. The authors should strive to answer my question about the impact of memory on the agent's performance.

---

> ### Author Response · Authors · 2022-11-15
> **Response to JNho**
>
> We thank the reviewer for their review. We are glad they found our paper “incredibly well written and detailed”, that “the experiments are thought provoking”, and “explores an interesting topic that should be interested to anyone ... working in the space of navigation”.
>
>
> > The agent is trained in such a way that the LSTM is expecting to receive all the previous observations from the start. As such, are the experiments providing out-of-distribution signals to the agent? Can the authors comment on the possibility of poor performance being a result of this shift in the nature of the data. Similarly, if the agent were retrained with a time-limited memory horizon (e.g., 100 or 500 steps), would we expect an increase in performance...
>
> Great question! We wondered this too and explored it in Section C.6 (on page 20), with results shown in Fig A14 (page 24). To summarize, we found that performance is better if the agent is trained to operate at that memory length, but still worse than unbounded memory length (up to a budget of 256). Due to computational constraints we weren’t able to train for lengths longer than 256 (training a memory length of 256 is 256x more computational expensive) and thus we can’t say when bounded memory would reach the performance of unbounded memory.
>
> Overall, the results in the main paper are accurate and robust in the claim that the agent is highly dependent on memory and that its memory has long-range dependencies, however they are imprecise on the exact length of those dependencies.
>
> We also note that while the agent was trained with an unbounded LSTM, it was trained with truncated back-prop-through-time (TBPTT). The dependencies learned certainly exceed the 256 truncation length used for training.
>
> We have updated the paper to clarify this point.
>
> > Relatedly, it would be helpful to have a separate curve (perhaps on the same plot) showing the success rate versus memory length in Fig. 2, as I believe this to be a key detail for understanding the performance with limited memory.
>
> Happy to add this. We have updated Fig. 2. Success largely follows the same trend as SPL. Indicating that as the memory budget increases, the agent is able to complete longer episodes but not complete existing ones more efficiently.
>
> > The line in Sec. 2 "but also that collisions-vs-not are linearly separable in this space": I do not believe that this claim logically follows from the previous clause. It may be true, but seems disconnected.
>
> We believe the claim
>
> - collisions-vs-not is linearly separate in the space (h_t+1, c_t+1)
>
> logically follows from the previous clause
>
> - a linear classifier evaluated on held-out data achieves 98% class-balanced accuracy (chance: 50%) on the task of predicting from (h_t+1, c_t+1) if a_t resulted in a collusion
>
> because the classifier is linear and we do not propagate gradients from the classifier to the agent's representations (i.e., we use frozen agent representations). To rephrase, linear separability follows from high empirical performance of a linear classifier.
>
> But perhaps we misunderstood a subtlety and the reviewer could clarify more?
>
> > Writing feedback and related work.
>
> Thank you for the feedback and pointers. We will work on incorporating it.
>
> Please let us know if you have any additional questions.
>
> Finally, if you believe this paper should be highlighted at the conference, could you please consider raising your score to reflect that?

---

### Official Review · Reviewer_XZvt · 2022-10-25

**Confidence:** 4
**Correctness:** 4
**Technical Novelty And Significance:** 4
**Empirical Novelty And Significance:** 3
**Recommendation:** 10

**Clarity, Quality, Novelty And Reproducibility:**

Clarity: high
Quality: high
Novelty: high
Reproducibility: Although there is no source code yet, the Appendix and the paper fully explains the training/experiment procedures, so it's medium.

**Strength And Weaknesses:**

- The strength of the paper comes from the novelty of the questions authors are asking. “Can blind agent navigate as well as non-blind agents?” lead to the bold, harsh conditions that agents are only given ego-motion sensors, without vision or any other sensor modalities.
- The results are meaningful that it is possible to efficiently navigate with just ego-motion sensing. The strong performance by blind agents are explained by the role of memory, and the memory encodes the implicit maps about the environment.
- In the section 2, the authors explain how the blind agents are becoming effective navigators (e.g. emergence of wall-following behavior and collision-detection neurons). In the section 3, the authors investigate how memory is utilized in the long-horizon settings.
- Memory also enables shortcuts, shown by the probe agent that is given the original agent’s memory containing spatial representations of the environment. The probe agent leverages the memory to enable shortcuts.
- I don’t see any weak points about the paper.

**Summary Of The Paper:**

While animal navigation assumes that there should be internal spatial representations (internal mental map) built and maintained for navigation, recent machine learning community’s works show that neural network-based models for navigation do not need explicit mapping modules; indeed, many models that do not contain explicit mapping achieve good performance in the exploration/navigation problems. This paper investigates a question “is mapping an emergent phenomenon?” In this work, the authors make their AI agent “blind”, which means that only egomotion sensor is present, making the agent having minimal perceptual system. There is no visual, audio, or any other sensing available to the agent. The paper argues that, under this harsh condition, the emergence of map-like spatial representations in the agent’s unstructured memory.


**Summary Of The Review:**

I would give acceptance score to this paper, although i'm open to changes after the discussion with other reviewers.

---

> ### Author Response · Authors · 2022-11-15
> **Response to XZvt**
>
>
> We thank the reviewer for their review. We are glad they found the questions we ask to be novel, our results to be meaningful, and that they “do see any weak points about the paper.”
>
> >  Reproducibility: Although there is no source code yet, the Appendix and the paper fully explains the training/experiment procedures, so it's medium.
>
> We will release source code upon publication.
>
> Please let us know if you have any additional questions.

---

### Comment · Program_Chairs · 2023-03-21
**Congratulations on winning the ICLR 2023 Outstanding Paper Award!!**

Hello authors,

Big congratulations! For more details about the award this year, please see https://blog.iclr.cc/2023/03/21/announcing-the-iclr-2023-outstanding-paper-award-recipients/

Best,

Mengdi, on behalf of ICLR 2023 PCs

---

### Public Comment · ~Michael_Maire1 · 2023-04-28
**prior work, spatial supervision, implicit curricula, and dissolving "surprise" (1/2)**

During the review process, in comments visible below, the Area Chair raised an issue of the novelty of this paper's central observation with respect to the results of Huynh et al. [1] --- a connection that was missed by the reviewers and authors, and not cited in the original submission.  As a co-author of [1], I am writing to clarify how specific experiments in that prior work relate to this paper, as well as dispute some claims of the latter.  The author response to the AC's comments mischaracterizes aspects of both prior work and their own paper.  Huynh et al. has a different aim and scope --- designing a new neural memory architecture and comparing the capabilities of spatial vs non-spatial memory.  However, a baseline experiment in Huynh et al. demonstrates that a blind agent with a non-spatial memory can learn to map a synthetic 2D maze.

This baseline experiment implements the agent as a Differentiable Neural Computer (DNC) [2] with a slot-based linear memory geometry, which does not match the 2D geometry of the environment.  Yet, at test time, the trained DNC-based agent is able to move through a novel 2D maze, update its memory from only local observations at each timestep, and answer queries about the structure of the maze it observed long ago --- effectively assembling a map of the 2D environment within the memory of the DNC.  The debate between the authors and AC below hinges upon whether the training setup of Huynh et al. is comparable to that of their paper.

A key point is the meaning of "spatial supervision" in the question "do spatial representations emerge in non-spatial neural networks without spatial supervision as a natural consequence of learning to navigate?"  However, the experiments in this paper (Wijmans et al.) do not adhere to the criterion of "without spatial supervision".  Moreover, both this paper and Huynh et al. utilize qualitatively similar kinds of spatial information when training their agents: relative GPS coordinates and the difference in spatial location between the agent and a reference.  This qualitative similarity is perhaps obscured by the difference in training regime between this paper (RL-based) and Huynh et al. (supervised pre-training, with later experiments adding exploration goals via RL), but can be elucidated as follows:

* (A) Wijmans et al. (this paper) states in Appendix A.1 that at each timestep, the agent receives the location of a goal (reference point), and its own GPS position and heading, all relative to the start location.  It receives a reward at every timestep based on the change in distance to the goal (A.1, Eq. 1).  The inputs to the agent, as well as the reward, explicitly depend upon the spatial coordinates of the agent and the goal.

    There is feedback at every timestep (dense supervision in time) that encourages the agent to move closer to the fixed reference point (goal).  Such a dense feedback signal provides a natural curriculum for training an RNN to build a map: remembering the closely surrounding environment will help in choosing immediately beneficial actions, and memory of larger portions of the environment will accrue benefits over longer time horizons.


* (B) Huynh et al. pre-train their agent by, at every timestep, giving the agent its GPS coordinates relative to the start, and querying it about a piece of maze structure (pattern of walls and free space).  The agent must respond by reporting the spatial locations (relative to start) at which it has previously encountered that structure.  Here, the reference point (location of query) changes at every timestep, but the agent must solve a similar problem involving its location and the reference.  Huynh et al. ask the agent to predict the relative location of the query pattern, whereas Wijmans et al. ask the agent to predict how to move closer to a fixed reference point.

    Again, dense feedback at every timestep provides a natural curriculum for training an RNN to build a map: remembering some previous observations (and accounting for relative motion) allows the agent to answer some queries, while remembering larger portions of the environment allows the agent to correctly answer more queries.

The RL vs supervised difference between (A) and (B) is a distraction from the fact that both papers engineer spatial supervision into every timestep of training the RNNs that comprise their agents.  It is objectively not surprising that an RNN will attempt to approximate a map when you train it with a dense feedback signal over time that encourages it to do precisely that, and moreover benefit from even incrementally learning to do so!

---

> ### Public Comment · ~Michael_Maire1 · 2023-04-28
> **prior work, spatial supervision, implicit curricula, and dissolving "surprise" (2/2)**
>
> Regarding the "emergence of spatial representations in non-spatial or unstructured neural networks", both this paper and Huynh et al. demonstrate that the geometry of the network architecture need not match the geometry of the environment.  Again, it is the DNC baseline experiment in Huynh et al. that demonstrates this point.  However, neither paper is justified in claiming this as a contribution.  Evidence that network architecture need not match task geometry predates both works.  For example, the DNC paper itself [2] shows that the DNC can ingest a graph and then answer shortest-path queries over that graph; the graph geometry does not match the DNC memory bank geometry.  Numerous recent works illustrate an even greater variety of designs which decouple network architecture from the spatial geometry of the input data or task.  Consider, for example, NeRFs [3] and Sirens [4].
>
> Finally, though "non-spatial or unstructured" networks are capable of learning spatial navigation tasks, Huynh et al.'s full set of experiments push back on any temptation to conclude that unstructured networks are the best choice.  Controlling for capacity, Huynh et al.'s spatial Multigrid memory architecture outperforms the non-spatial DNC on spatial tasks, matches it on non-spatial tasks, and has unique advantages on some hybrid tasks.  Spatially-structured neural memory might actually be the more general-purpose design.
>
>
> References:
>
> 1. Tri Huynh, Michael Maire, and Matthew R. Walter, "Multigrid Neural Memory" ICML, 2020.
>
> 2. Alex Graves et al., "Hybrid Computing using a Neural Network with Dynamic External Memory" Nature, 2016.
>
> 3. Ben Mildenhall, Pratul P. Srinivasan, Matthew Tancik, Jonathan T. Barron, Ravi Ramamoorthi, Ren Ng, "Representing Scenes as Neural Radiance Fields for View Synthesis" ECCV, 2020.
>
> 4. Vincent Sitzmann, Julien N. P. Martel, Alexander W. Bergman, David B. Lindell, Gordon Wetzstein, "Implicit Neural Representations with Periodic Activation Functions" NeurIPS, 2020.

---

> > ### Author Response · Authors · 2023-05-09
> > **Unclear set of comments in (2/2)**
> >
> > > Regarding the "emergence of spatial representations in non-spatial or unstructured neural networks", both this paper and Huynh et al. demonstrate that the geometry of the network architecture need not match the geometry of the environment. Again, it is the DNC baseline experiment in Huynh et al. that demonstrates this point. However, neither paper is justified in claiming this as a contribution.
> >
> > Our manuscript does not claim the contribution you quote because your quote makes no mention of navigation (as we explain above, that is critical). A full list of our contributions is available in our manuscript intro. We are happy to answer any questions about those contributions.
> >
> > > Finally, though "non-spatial or unstructured" networks are capable of learning spatial navigation tasks, Huynh et al.'s full set of experiments push back on any temptation to conclude that unstructured networks are the best choice.
> >
> > We are unsure why you bring this up. Our manuscript makes no recommendations or claims about unstructured networks being the best choice for navigation.
> >
> > Perhaps you are simply trying to leave thoughts and recommendations for future readers who may be lacking the full context of the literature. If so, great.

---

> ### Author Response · Authors · 2023-05-08
> **Let's keep our sights on the question our paper is asking**
>
> Hi Michael,
>
> Thanks for contributing to this discussion; we appreciate you taking the time.
>
> > The author response to the AC's comments mischaracterizes aspects of both prior work and their own paper.
>
> Before we respond to the specific questions/comments below, we want to note that:
> 1. We view the manuscript as the primary artifact. This discussion here is merely in service of creating this artifact.
> 2. We are happy to make changes to the manuscript to provide future readers with an accurate understanding.
>
> In that spirit, could you please tell us if you believe our _manuscript_ mischaracterizes anything and if so, how?
>
> > A key point is the meaning of "spatial supervision" in the question "do spatial representations emerge in non-spatial neural networks without spatial supervision as a natural consequence of learning to navigate?"
>
> It’s easy to get lost in the weeds in this back-and-forth dialog, so let us re-focus the attention back on our manuscript intro.
> - We investigate the following question: do AI agents learn to build internal spatial representations (or ‘mental’ maps) of their environment as a natural consequence of learning to navigate?
> - In animals, navigation is supported by a number of mechanisms (landmark localization, olfaction, etc). We design our study to render such alternative mechanisms implausible.
> - Since we are asking “does X emerge in the course of learning Y”, we must: (a) train for Y, and (b) take away the inductive bias for X (to the degree possible).
> - This is what gets operationalized as: "do spatial representations emerge in non-spatial neural networks without spatial supervision as a natural consequence of learning to navigate?"
>
> In order to address these questions, we must start by training a navigation method (withholding the inductive bias for mapping to the degree possible) and check for the emergence of maps. All methods in [1] (including the DNC baseline) operate in the opposite direction – they first pre-train a mapping and localization network, then freeze this network and train a navigation (exploration) policy on top. And to be clear, this is not a mistake in [1], it is (as you also agree) addressing a different goal with a different scope.
>
> > However, the experiments in this paper (Wijmans et al.) do not adhere to the criterion of "without spatial supervision".
>
> We believe you are missing our goal.
>
> Recall that our goal is: do AI agents learn to build internal spatial representations (or ‘mental’ maps) of their environment as a natural consequence of learning to navigate?
>
> We must train for navigation otherwise we are not addressing this question. There _must be some spatial supervision_ because the navigation problem we study is a spatial navigation problem (as opposed to, say, navigating an undirected/directed graph like the internet).
>
> A version of the operationalized question that makes this clearer is “do spatial representations emerge in non-spatial neural networks as a natural consequence of learning to navigate and without spatial supervision **other than navigation**?”
>
> > This baseline experiment implements the agent as a Differentiable Neural Computer (DNC) [2] with a slot-based linear memory geometry, which does not match the 2D geometry of the environment. Yet, at test time, the trained DNC-based agent is able to move through a novel 2D maze, update its memory from only local observations at each timestep, and answer queries about the structure of the maze it observed long ago
>
> We presume you are referring to Table 1 rows 2 and 4 (corresponding to DNC). As [1] states in Section 4.1, those results report “performance in terms of localization accuracy…in which the agent moves in a spiral”. The only navigation results (on exploration, not PointNav) are reported in Fig 5 for the multigrid memory (not DNC).
>
> In fact, as far as we can tell, _no result is presented in [1] anywhere_ for a navigation policy with DNC. It is only evaluated for localization.
>
> So we are unsure what your claim “trained DNC-based agent is able to move through a novel 2D maze” is based on.
>
> Perhaps you meant to say: “a DNC-based mapper is able to localize itself when moved along a fixed spiral trajectory”. If so, that is a very different claim; an interesting one, but one that doesn’t address the question our paper is studying.

---

### Decision · Program_Chairs · 2023-01-20

**Decision:**

Accept: notable-top-5%

**Justification For Why Not Higher Score:**

N/A

**Justification For Why Not Lower Score:**

None. This paper is worthy of at least a spotlight.

**Metareview: Summary, Strengths And Weaknesses:**

The paper studies the emergence of implicit maps within standard recurrent architectures trained to navigate a "blind" agent that has access only to proprioceptive (egomotion) observations. The paper describes an extensive set of experiments that investigate the nature of the agent's implicit learned memory and the ability with which it enables the "blind" agents to navigate, and analyzes the spatial information modeled in this memory.
The paper was very well received by all four reviewers, with overall ratings that include two accepts (8) and two strong accepts (10). There is unanimous agreement among the reviewers that the paper is incredibly well written and highly thought provoking. As readily acknowledged by the authors, the contribution is not in proposing new navigation algorithms, but rather in presenting a meaningful investigation of the emergence of spatial memory in enabling navigation through a comprehensive set of results.
We agree with the reviewers regarding the interesting insight that the paper provides into learned spatial memory. However, it would be more comprehensive to discuss some related prior work; some examples below:

T Huynh, M Maire, MR Walter. Multigrid Neural Memory. ICML 2020

Graves, Alex, Greg Wayne, Malcolm Reynolds, Tim Harley, Ivo Danihelka, Agnieszka Grabska-Barwińska, Sergio Gómez Colmenarejo et al. "Hybrid computing using a neural network with dynamic external memory." Nature 538, no. 7626 (2016): 471-476.


**Note From Pc:**

if the above contains the word "oral" or "spotlight" please see: "oral" presentation means -> notable-top-5% and "spotlight" means -> notable-top-25%. As stated in our emails, we are disassociating presentation type from AC recommendations

**Summary Of Ac-Reviewer Meeting:**

N/A

---

> ### Author Response · Authors · 2023-01-21
> **Differences to Huynh et al.**
>
> We thank the AC for bringing the work of Huynh et al. to our attention. This is a related work, and we will add it to the manuscript.
>
> However, there appears to be a fundamental misunderstanding here:
> - Our work demonstrates the emergence of spatial representations in *non-spatial* or unstructured neural networks *without any inductive bias towards mapping*.
> - Hunynh et al. propose a *spatial memory explicitly trained with a localization loss*.
>
> The two works are operating in different experimental conditions and addressing different questions. These differences make our findings a priori unknown and a posteriori surprising.
>
> We describe these differences in detail below.
>
> Authors
>
> -----
>
> 1) *Architectural Differences.*
> Huynh et al. propose “Multigrid neural memory”, an explicitly spatial memory architecture. Huynh et al. state that mapping and localization is an “inherently spatial task” and “naturally fit to the architecturally-induced biases of [their] memory network”. Our work uses task-agnostic neural components (LSTMs and MLPs), which do not contain any explicit spatial components.
>
> 2) *Supervision Differences.*
> Huynh et al. explicitly train the mapping component of their network for a mapping+localization task. Specifically, they supervise the network with a localization loss to label all previously seen locations of a given patch of a 2D grid (environment). In contrast, the only learning signal provided in our experimental setup is the navigational reward. We do not back-propagate gradients from the probes (referred to as “readers” in Huynh et al.) into the neural memory; they are solely for diagnostic purposes.
>
> Cumulatively, the experimental conditions in Huynh et al. involve creating a network with “architecturally-induced biases” towards mapping and training for localization, not navigation.
>
> Our work studies the question “do spatial representations emerge in task-agnostic (non-spatial) neural networks without spatial supervision as a natural consequence of learning to navigate?” The work of Huynh et al. does not provide an answer to this question.

---

> > ### Comment · Area_Chair_BPnk · 2023-01-27
> > **Re: Differences to Huynh et al.**
> >
> > I thank the authors for following up regarding the relationship to the work of Huynh et al.
> >
> > As the authors point out, Huynh et al. use an architecture that is particularly well suited to tasks such as navigation that involve spatial reasoning, whereas the paper under consideration demonstrates that an architecture without such an inductive bias is capable of learning spatial information. The results of Huynh et al., involve the use of an auxiliary localization objective, which one can reasonably expect would only encourage the emergence of spatial maps. In the case of the paper under consideration here, the goal distance-based reward also provides a form of spatial supervision, albeit one that is arguably less explicit. It doesn't seem surprising that this, coupled with the fact that the agent is "blind", encourages the model to learn spatial information.
> >
> >     Our work studies the question “do spatial representations emerge in task-agnostic (non-spatial) neural networks without spatial supervision as a natural consequence of learning to navigate?” The work of Huynh et al. does not provide an answer to this questio
> >
> > A baseline experiment in Huynh et al. directly answers this question in the positive.  They show that the Differentiable Neural Computer (DNC), which has a linear slot-based memory rather than a 2D spatial memory, achieves nontrivial accuracy on the following task: Given a local observation at each step in a sequence of local movements (e.g, 100 steps) in a 2D maze, answer questions about the 2D structure of the environment that has been explored (including pieces seen far in the past).  This result is in the second row of Table 1 of Huynh et al.
> >
> > The fact that Huynh et al. also show the DNC (Graves et al., 2016), which does not have a 2D spatial memory architecture, is significantly outperformed by a Multigrid Memory network which does have such internal architecture, suggests there may be limits to the accuracy or learning efficiency we might expect from simpler architectures on spatial memory tasks.  But, it should not be surprising to have some degree of success with linear memory architectures.
> >
> > Graves, Alex, Greg Wayne, Malcolm Reynolds, Tim Harley, Ivo Danihelka, Agnieszka Grabska-Barwińska, Sergio Gómez Colmenarejo et al. "Hybrid computing using a neural network with dynamic external memory." Nature 538, no. 7626 (2016): 471-476.

---

> > > ### Author Response · Authors · 2023-02-03
> > > **Differences to Huynh et al. (Part 2)**
> > >
> > > We thank the AC for their response, however there still appears to be a misunderstanding.
> > >
> > > > The results of Huynh et al., involve the use of an auxiliary localization objective
> > >
> > > Not quite. Their localization objective is not an _auxiliary_ objective for their mapping results, but the _sole_ objective. The navigation policy trained in Huynh et al. is learned “on top of a fixed mapping and localization network pre-trained with random walk motion.” (Section 4.2) This may seem like a minor distinction but in this context, it is not – sole supervision means that it is not answering the question asked in our paper (quoted in AC’s response above).
> > >
> > > > A baseline experiment in Huynh et al. directly answers this question in the positive.
> > >
> > > It does not. The network in this experiment was trained for localization.
> > >
> > > We repeat the goal of our paper quoted by the AC with an emphasis added: "Do spatial representations emerge in non-spatial neural networks **without spatial supervision** as a natural consequence of learning to navigate?”
> > >
> > > > In the case of the paper under consideration here, the goal distance-based reward also provides a form of spatial supervision, albeit one that is arguably less explicit. It doesn't seem surprising that this, coupled with the fact that the agent is "blind", encourages the model to learn spatial information.
> > >
> > > “Surprise” is surprisingly hard to find in hindsight.
> > >
> > > Would it have been surprising had maps not emerged?
> > >
> > > We would argue that a lack of emergent maps would not be considered surprising, given that the deep learning and RL literature is replete with networks optimizing reward and metrics for the wrong reasons, enabling them to solve tasks via shortcuts. If lack of emergence wouldn’t have been surprising, then emergence is surprising. If neither is surprising, it’s likely us falling for hindsight bias.
> > >
> > > Finally, we also note that we are unaware of any prior work that actually formulated and tested our hypothesis. Testing and measuring previously untested hypotheses is a core importance of scientific advance and understanding.

---

> > > > ### Comment · Area_Chair_BPnk · 2023-02-11
> > > > **Meaning of Spatial Supervision**
> > > >
> > > > Quoting the authors above (emphasis their's):
> > > >
> > > > > We repeat the goal of our paper quoted by the AC with an emphasis added: "Do spatial representations emerge in non-spatial neural networks **without spatial supervision** as a natural consequence of learning to navigate?”
> > > >
> > > > I find it difficult to see how one could argue that the architecture is not provided with spatial supervision. The first paragraph in Section 2 includes the following statement
> > > >
> > > > > It [the agent] is equipped with an egomotion sensor providing it relative position $(\Delta x, \Delta y, \Delta z)$ and relative ‘heading’ (or yaw angle) $\Delta \theta$ between successive steps, which is integrated to keep track of the agent’s location and heading relative to start $[x_t, y_t, z_t, \theta_t]$. This is sometimes referred to as a 'GPS+Compass' sensor
> > > >
> > > > The phrasing is vague, but this can be interpreted as stating that the agent is provided with knowledge of either its precise absolute position $(x_t, y_t)$[^1] and heading $\theta_t$ relative to the start at each time step (consistent with the the sensor being labeled as a 'GPS+Compass' sensor[^2]) or, at the very least, the precise change in position $(\Delta x, \Delta y)$ and heading $\Delta \theta$ between timesteps, the latter providing implicit knowledge of the agent's absolute position and heading at every timestep. It is difficult to understand how this explicit or implicit 'GPS+Compass' supervision is anything but spatial supervision, particularly since perfect (i.e., noise-free) egomotion sensors do not exist (e.g., egomotion sensors are the primary source of noise in robot mapping, localization, and SLAM).
> > > >
> > > > Huynh et al., on the other hand, do not provide the network with the agent's position or heading (neither relative nor absolute). Instead, as the authors point out, Huynh et al., train the network with a loss that encourages it to remember a location (position) that it previously visited. If anything, this seems like a weaker form of spatial supervision than is provided here.
> > > >
> > > > [^1] Note: I am deliberately omitting $z_t$
> > > >
> > > > [^2] A GPS and compass provide absolute observations of position and heading, respectively, hence the confusion as to whether the supervision is in the form of relative or absolute position and orientation.

---

> > > > > ### Author Response · Authors · 2023-02-16
> > > > > **Re: Meaning of Spatial Supervision**
> > > > >
> > > > > Thanks for the continued engagement. Happy to answer these questions.
> > > > >
> > > > > > Huynh et al., on the other hand, do not provide the network with the agent's position or heading (neither relative nor absolute).
> > > > >
> > > > > Not accurate. Huynh et al. provide the agent’s memory “writer” with “[the agent’s] location relative to the starting point” (See section B.1.1, also see Fig 11 and its caption). The heading is inapplicable in their experiment because the agent cannot turn in their 2D grid-world.
> > > > >
> > > > > > Is “the agent provided with … absolute position and heading relative to the start [or] the .. change?
> > > > >
> > > > > Absolute (relative to start). As described in Section 2 bottom para (a few lines below the one quoted by the AC), the input to the policy is o_t = [x_g , y_g , z_g , x_t, y_t, z_t, θ_t], and x_t, y_t, z_t, theta_t are defined in the lines quoted by the AC.
> > > > >
> > > > > > meaning of supervision
> > > > >
> > > > > By supervision, we mean supervision as generally used by the machine learning community. The sentences quoted describe the sensors of our agent, which is the input to the policy and is typically not considered supervision.
> > > > > (This is ignoring complications brought by the phrase “self-supervised learning”, which we do not use.)
> > > > >
> > > > > We agree that there can be nuanced and subtle takes on “what is the meaning of supervision in ML”, but that is beyond the scope of this paper.

---

> > > > > > ### Comment · Area_Chair_BPnk · 2023-03-29
> > > > > > **Re: Meaning of Spatial Supervision**
> > > > > >
> > > > > > Apologies for not responding sooner. I only saw this the other day (ideally, I wouldn't have to click on "View 1 More Reply" for OpenReview to render all comments).
> > > > > >
> > > > > > Thank you for pointing out the the discrepancy with my statement about Huynh et al. While the policy is not provided with the agent's location, you are correct that it is provided to the "writer".
> > > > > >
> > > > > > > By supervision, we mean supervision as generally used by the machine learning community. The sentences quoted describe the sensors of our agent, which is the input to the policy and is typically not considered supervision. (This is ignoring complications brought by the phrase “self-supervised learning”, which we do not use.)
> > > > > >
> > > > > > > We agree that there can be nuanced and subtle takes on “what is the meaning of supervision in ML”, but that is beyond the scope of this paper.
> > > > > >
> > > > > > My point is that it is misleading to say that the architecture is not provided with localization supervision (Section 7: "In contrast, our work provides no direct supervision for path integration, localization, or mapping") simply because the absolute, noise-free pose is *explicitly* given to the agent as opposed to being incorporated into the loss/reward. I don't think that this involves a nuanced take on the meaning of supervision.
> > > > > >
> > > > > > Indeed, recent work [1] shows that the assumption that an agent has access to perfect Compass+GPS absolute pose is "unrealistic" (as has long been known in robotics), and that when this assumption can not be met, agents that otherwise achieve high success on PointGoal navigation demonstrate a significant decrease in performance (from 99.6% to 0.3% in their study). Of course, this is not to say that navigation and related tasks **require** access to perfect pose information---indeed, decades of work in "probabilistic robotics" has demonstrated the utility of fusing multiple sources of noisy supervision for these tasks, while [1] nicely shows that the use of visual odometry can make up for the lack of perfect Compass+GPS pose.
> > > > > >
> > > > > > To be clear, I believe that this paper provides an interesting investigation into the spatial awareness capabilities of architectures that are not explicitly designed to have such an inductive bias. I just think that it is important to not overstate the contributions.
> > > > > >
> > > > > > [1] Xiaoming Zhao, Harsh Agrawal, Dhruv Batra, and Alexander G. Schwing. "The surprising effectiveness of visual odometry techniques for embodied pointgoal navigation." ICCV 2021.
> > > > > > https://openaccess.thecvf.com/content/ICCV2021/papers/Zhao_The_Surprising_Effectiveness_of_Visual_Odometry_Techniques_for_Embodied_PointGoal_ICCV_2021_paper.pdf